# Both genome instability and replicative senescence stem from the shortest telomere in telomerase-negative cells

Prisca Berardi[1,9], Veronica Martinez-Fernandez[1,9], Anaïs Rat [2],
Fernando R. Rosas Bringas [3], Pascale Jolivet[1], Rachel Langston[1],
Stefano Mattarocci[1], Alexandre Maes[1], Théo Aspert [4,5,6,7], Bechara Zeinoun[1],
Karine Casier [1], Hinke G. Kazemier [3], Gilles Charvin [4,5,6,7], Marie Doumic[8],
Michael Chang [3] & Maria Teresa Teixeira [1] ✉

In the absence of telomerase, telomere shortening triggers replicative senescence, a tumor suppressor mechanism that is also associated with oncogenic genomic instability. Yet, the precise mechanism that connects these seemingly opposing forces remains poorly understood. To directly study the complex interplay between senescence, telomere dynamics, and genomic instability, we develop a system in *Saccharomyces cerevisiae* to generate and track telomeres of precise length in the absence of telomerase. Using single-telomere and single-cell analyses combined with mathematical modeling, we identify a threshold length at which telomeres switch into dysfunction. A single shortest telomere below the threshold length is necessary and sufficient to trigger the onset of replicative senescence in a majority of cells. At population level, fluctuation assays establish that rare genomic instability arises predominantly in *cis* to the shortest telomere as Pol32-dependent non-reciprocal translocations that result in re-elongation of the shortest telomere and likely transient escape from senescence. The switch of the shortest telomere into dysfunction and subsequent processing in telomerase-negative cells thus serves as the mechanistic link between replicative senescence onset, genomic instability and the initiation of post-senescence survival.

Telomeres, located at the ends of linear chromosomes in eukaryotes, comprise G/C-rich repeated DNA sequences–with the G-rich strand extending to form a 3′ protruding end, called G-overhang, as well as specialized proteins and non-coding RNA. The primary function of telomeres is to shield chromosome ends from being recognized as DNA extremities resulting from double-strand breaks (DSBs) and prevent DNA repair activities that might compromise genome stability[1,2]. During DNA replication, telomeres naturally undergo shortening in a phenomenon referred to as the DNA-end replication problem[3–5]. Telomerase, a specialized ribonucleoprotein complex, counteracts this shortening by synthesizing telomeric repeats de novo using its reverse transcriptase subunit and template RNA[6,7].

[1]Sorbonne Université, CNRS, Laboratoire de Biologie Moléculaire et Cellulaire des Eucaryotes, LBMCE, Paris, France. [2]Univ Brest, CNRS, UMR 6205, Laboratoire de Mathématiques de Bretagne Atlantique, Brest, France. [3]European Research Institute for the Biology of Ageing, University of Groningen, University Medical Center Groningen, Groningen, The Netherlands. [4]Department of Developmental Biology and Stem Cells, Institut de Génétique et de Biologie Moléculaire et Cellulaire, Illkirch, France. [5]CNRS, UMR7104, Illkirch, France. [6]INSERM, U964, Illkirch, France. [7]Université de Strasbourg, Illkirch, France. [8]CMAP, Inria, IP Paris, Ecole Polytechnique, CNRS, Palaiseau cedex, France. [9]These authors contributed equally: Prisca Berardi, Veronica Martinez-Fernandez. ✉e-mail: teresa.teixeira@cnrs.fr

Telomerase plays a crucial role in the long-term proliferation of eukaryotic cells. Nonetheless, in many somatic cells of humans and some other species, telomerase activity is repressed. In these cells, as well as in experimental model systems, inactivation of telomerase leads to telomere erosion, eventually causing replicative senescence— a permanent cell cycle arrest triggered by the activation of the DNA damage checkpoint[8–10]. For this reason, replicative senescence has been considered a potent tumor-suppressor mechanism. However, cancer cells often escape this mechanism through the acquisition of the ability to re-extend telomeres by either re-expressing telomerase or activating pathways collectively termed alternative lengthening of telomeres (ALT)[11,12]. ALT pathways, first observed in *Saccharomyces cerevisiae* cells lacking telomerase, involve homology-dependent repair mechanisms such as break-induced replication (BIR)[13–16]. Despite significant progress, the mechanisms by which short telomeres trigger replicative senescence and by which cells activate ALT pathways remain elusive, in part due to the lack of systems that allow direct testing of hypotheses.

Genomic instability also increases with telomere shortening. Accordingly, rare gross chromosomal rearrangements such as translocations, or more massive genomic catastrophes, have been attributed to presumably dysfunctional short telomeres[17–21]. Thus, when combined with genomic instability, ALT could theoretically enable the long-term proliferation of cells carrying genomic rearrangements that present a threat to organismal long-term survival. However, whether and how genomic instability is mechanistically linked to ALT is unknown. Moreover, the very nature of the initial telomere dysfunction at the origin of the genomic instability is unclear and difficult to assess. Several non-exclusive hypotheses have been put forward, including replication fork collapse when passing through hard-to-replicate telomeric repeated sequences, oxidation of telomeric DNA following mitochondrial dysfunction, and extreme telomere shortening[22,23]. However, none of these hypotheses provide a comprehensive explanation. Also, the study of replicative senescence presents challenges due to inherent variability across telomeres and cellular phenotypes associated with senescence[24]. Telomere length varies not only between different telomeres within a single cell but also among cells. Additionally, the telomere shortening mechanism hides several layers of complexity, which include the asymmetry of leading and lagging replication machineries contributing to variations in telomere length distributions[4,25]. Variability also stems from significant inter-cellular differences in proliferation potential and gene expression[26]. Furthermore, senescence-specific genomic instability contributes to increased genetic diversity, further complicating the study of the mechanisms involved in senescence[18,19]. Therefore, it is particularly challenging to directly connect a specific molecular event at specific telomere(s) to the onset of senescence.

The heterogeneity observed in telomere biology is recapitulated in the budding yeast *Saccharomyces cerevisiae*, where telomeres consist of degenerated $TG_{1-3}$ repeats spanning approximately $300 \pm 75$ bp[27,28]. Mutant yeast cells that lack telomerase experience gradual asymmetrical telomere shortening at an average rate of -3–5 bp per cell division[4,29]. As telomeres shorten, an increasing but highly variable fraction of cells undergo replicative senescence[9,30]. Studies in this organism suggest a model in which telomerase-negative budding yeast cells arrest and activate DNA damage checkpoints as soon as one telomere reaches a critically short length[22,31–33]. Notably, microscopy tracking in microfluidics circuits of consecutive cell divisions over time, from telomerase inactivation to cell death, combined with mathematical modeling, have provided valuable insights into the intricate relationship between telomere shortening and replicative senescence in budding yeast telomerase-negative cells[25,30,34–37]. However, the hypothesis that a single telomere is sufficient to trigger senescence remains controversial and was not possible to be directly tested in a physiological system with native telomeres. In order to understand the mechanisms underlying the onset of replicative senescence and its escape, it is thus crucial to examine the structure and fate of the shortest telomere in budding yeast cells lacking telomerase.

In this work, we present a system called *FinalCut*, which enables the generation in budding yeast cells of a single short telomere with a defined length to control telomere length heterogeneity. *FinalCut* targets the ectopic endonuclease Cas9 to a short sequence within the $TG_{1-3}$ repeats of a specific native telomere. Combining structural analysis of the rate of *FinalCut* telomere shortening by single-molecule sequencing, together with the monitoring of single consecutive cell cycles and mathematical modeling, we show that the shortest telomere in cells reaching a critical length threshold is required and sufficient to trigger replicative senescence in a large majority of telomerase-negative cells. Our data also demonstrate that genome instability upon telomerase inactivation occurs primarily in the vicinity of the critically short telomere, while the stability of the rest of the genome does not appear to be compromised. Furthermore, non-reciprocal translocations appear as a mutational signature associated with the shortest telomere when telomerase is absent. Overall, our findings demonstrate that the shortest telomere in cells is both the primary cause of proliferation limit and the major threat to genome integrity.

## Results

### The *FinalCut* system enables precise experimental control of telomere length

While telomere length appears as a critical determinant of telomere biology, telomere length variation often masks the understanding of underlying molecular mechanisms. Therefore, the precise experimental control of telomere length is a challenging but essential objective. Experimental systems that aim at modifying telomere length exist, but they exhibit limitations. They often involve alterations of subtelomeric elements, exposure of non-telomeric DNA ends, or lack precision in achieving the desired final telomere length. We thus undertook the development of a tool we named *FinalCut*. By utilizing an inducible CRISPR/Cas9 approach, the *FinalCut* allows us to selectively truncate a single native telomere to a precise length with temporal control (Fig. 1a).

To ensure a controlled Cas9 cleavage of the telomere, we chose the right end of the chromosome 6 (*TEL6R*) to integrate a construct containing a variable number of native 6R telomeric $TG_{1-3}$ repeats, a unique 29 bp ectopic sequence containing a protospacer adjacent motif sequence and Cas9 guide RNA-complementary target sequence (ttDNA), and a set of terminal $TG_{1-3}$ repeats. *TEL6R* location was chosen because of the well-characterized nature of this telomere[38,39]. In the integrated context, the variable amount of telomeric sequence before the Cas9 binding site determines the length of the telomere after the Cas9 cut. To enable precise interrogation of the impact of telomere length, we generated a set of strains with *TEL6R* lengths of 0, 21, 31, 41, 51, or 70 bp of the native $TG_{1-3}$ (*FCx*), which we named *FC0*, *FC20*, *FC30*, *FC40*, *FC50*, *FC70*, respectively (*noFC* refers to the original control strain harboring an untouched *TEL6R*).

To temporally control Cas9 cleavage, we use a plasmid-based galactose-inducible Cas9 enzyme. Further control of the telomere was achieved by constructing the *FinalCut* system in strains that placed the telomerase template RNA encoding gene, *TLC1*, under the control of a doxycycline repressible promoter ($P_{tetO2}$-*TLC1*) so that telomerase could be deactivated as appropriate[4].

We collected DNA samples from different time points of galactose induction of the *FinalCut* strains and analyzed by Southern blots and TeloPCR[40] for detection of the terminal restriction fragment (TRF) of the 6 R chromosome end (Fig. 1b and Supplementary Fig. 1a). At $t = 0$ prior to galactose induction, a smeary signal was identified, showing

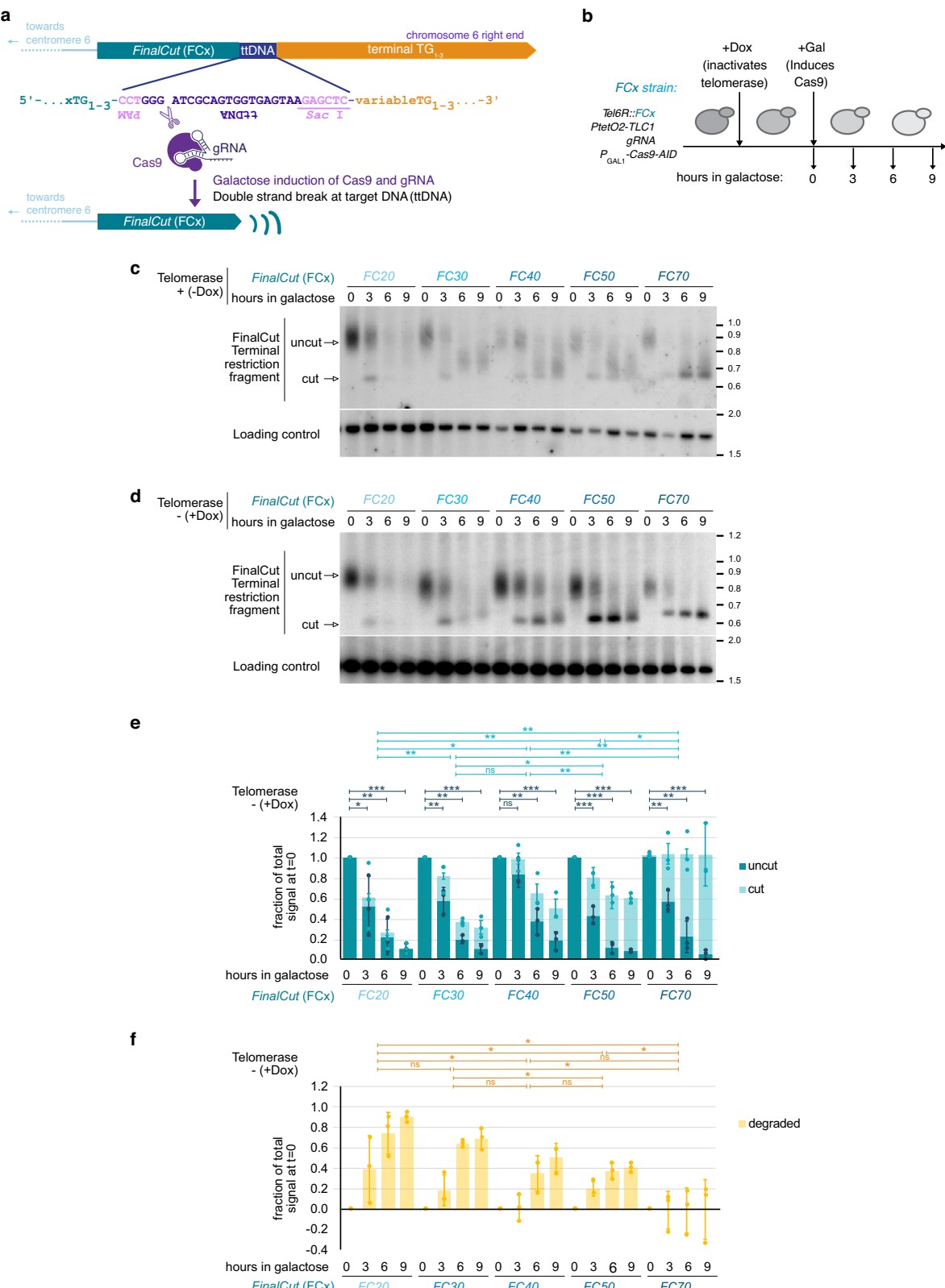

that the *FinalCut* construct is well maintained by telomerase to a variable length (Fig. 1c, d and Supplementary Fig. 1b and f), as described[4,41]. Upon galactose induction, this signal was converted into a sharp band both in the presence and absence of telomerase activity. This demonstrated efficient cleavage of the *FinalCut* constructs by Cas9 to produce telomeres at 6 R of the expected lengths (Fig. 1e). Detection of all telomeres using a $CA_{1-3}$ probe showed that the

*FinalCut* telomeres were the only ones cut in these conditions (Supplementary Fig. 1c).

In the presence of telomerase, we observed that in the *FC30-70* strains, the Cas9-shortened 6R telomere fragments were rapidly elongated to longer fragments, whereas the signal in the *FC20* strain vanished (Fig. 1c). This likely reflects the ability of telomeric sequences above a threshold of ~30 bp to be elongated by telomerase[42], whereas

**Fig. 1 | The *FinalCut* system efficiently generates a unique telomere of defined length. a** The *FinalCut* construct at telomere 6R: upon induction of Cas9 and association with a specific guide RNA (gRNA), a single telomere in cell containing the complementary sequence of gRNA, ttDNA, is cut to generate a telomere of defined length FCx. **b** Experimental procedure. Strains harbouring a telomere 6R modified to display the *FinalCut* constructs, are precultured in Raffinose media containing or not Doxycycline to turn off the promoter controlling telomerase template RNA gene ($P_{tetO2}$-*TLC1*). Galactose is then added to induce Cas9 gene ($P_{GAL1}$-*Cas9*). Guide RNA is expressed constitutively from gDNA locus. Samples are taken at indicated time points. **c**, **d** DNA is prepared at indicated time points from strains with indicated *FinalCut* telomere lengths in the presence (**c**) or absence (**d**) of telomerase, digested with *Hpa* I restriction enzyme to generate 6R terminal restriction fragments, electrophoresed and probed with the subtelomere 6R to reveal the *FinalCut* terminal restriction fragment or an internal locus in chromosome 11 (Loading control). **e** Quantification of three independent experiments as in (**d**). "Uncut" and "cut" bands normalized to loading control and to "uncut" signal

measured at $t = 0$ h. Small symbols: values from independent experiments. Bars correspond to average and error bars to SD of three independent experiments. Statistical differences between 0 h and remaining time points of "uncut" were determined by two-sided Student $t$-tests (dark green lines). Differences in the decay kinetics of the "uncut" among the *FinalCut* constructs were evaluated by comparing the slopes of linear regressions using a one-way ANOVA ($p$-value = 0.44, not depicted). Differences in the stability of the "cut" over time among the *FinalCut* constructs were evaluated by comparing the slopes of linear regressions with two-sided Student's $t$-tests (light green lines). *: $p < 0.05$; **: $p < 0.005$; ***: $p < 0.0005$; ns not significant. **f** Quantification of the disappearance of the signal from the Southern blot membranes of three independent experiments. Bars correspond to average and error bars to SD of three independent experiments. Differences in the "degraded" over time among the *FinalCut* constructs were evaluated by comparing the slopes of linear regressions with two-sided Student's $t$-tests (yellow lines). *: $p < 0.05$; ns not significant. Source data are provided as a Source Data file.

below, they are rapidly degraded. We also observed a relative decrease in the quantity of telomerase-elongated *FC70* telomeres, since telomerase is less frequently recruited to relatively longer telomeres[43]. These observations suggest that the *FinalCut* telomeres are functional with respect to telomerase recruitment.

When telomerase was inactivated prior to Cas9 induction by doxycycline addition to the media to suppress *TLC1* expression, approximately 80–90% of the uncut smear signal was converted to a sharp cut band within 6–9 h (Fig. 1d, e, Supplementary Fig. 1d and Supplementary Table 6). We also found that the total signal intensity of *FinalCut* telomeres gradually decrease over time, with a more pronounced effect on shorter telomeres compared to longer telomeres. Indeed, while we observed a rapid disappearance of the 6R TRF signal in both *FC20* and *FC30* strains, the signal persisted for *FC40-70* strains, when the cut telomere was 40 bp or longer (Fig. 1f). Since no other signal at other lengths was detected (Supplementary Fig. 1b), we could exclude the possibility of a systematic fusion or recombination of shorter *FinalCut* telomeres with another specific region of the genome. A minor subset of *FinalCut* telomeres above 30 bp exhibited lengthening. This phenomenon likely arises from sporadic $P_{tetO2}$ leakage in a subset of cells and selection, given that no such elongations were observed in the *FC20* strain, as expected for telomerase not recognizing a 20 bp-telomeric sequence as a substrate[42]. Overall, our results indicate that most of the shorter *FinalCut* telomeres of *FC20* and *FC30* strains are likely degraded immediately, whereas the ones of *FC40* and *FC50* are initially stable prior to delayed degradation (Fig. 1f). Therefore, while telomerase can re-elongate telomeres above 30 bp, as previously described[42], we find that telomeres shorter than 40 bp in telomerase-negative cells become susceptible to degradation. This suggests that telomere structures enabling telomerase recruitment and telomere protection might differ substantially.

Probing at a more internal restriction fragment, 23 kb away from *TEL6R*, indicates that degradation at *FC20* and *FC30* may extend internally into chromosomal regions containing essential genes (Supplementary Fig. 1d, e). We propose that the shift between telomere function to dysfunction manifests as end degradation progressing toward the interior of the chromosome. The structural nature of this shift may have remained directly undetected likely because it affects only a small fraction of the 32 telomeres in the budding yeast genome, as illustrated in the Supplementary Fig. 1c, and a small fraction of cells in a population[30].

### Temporal control of *FinalCut* telomere generation reveals threshold for telomere dysfunction

To further improve the *FinalCut* system to enable evaluation of telomere stability, we decided to temporally restrain the generation of *FinalCut* telomeres before evaluating their stability over time. This eliminates the confounding variable of *FinalCut* telomeres being

continuously generated as cells divide and telomeres shorten and/or degrade. In this enhanced system, we provided galactose for 4 h, a period shown to be sufficient to allow a substantial production of *FinalCut* telomeres, and then changed the carbon source to glucose to repress the *GAL1* promoter driving Cas9 expression (Fig. 2a). Additionally, to promote rapid degradation of the remaining Cas9, we used a version of the nuclease fused to the auxin-inducible degron (AID) and induced rapid degradation of Cas9 by addition of the auxin analogue 1-Naphthalenacetic acid (NAA) (Supplementary Fig. 2a-c)[44,45].

Intriguingly, analysis of *FinalCut* TRFs across different time points upon Cas9 shut off showed that a telomere of 30 bp is rapidly degraded after the cut and hardly quantifiable (Fig. 2b, c and Supplementary Fig. 2d). In contrast, the signal did not decrease in the case of a 40 bp or a 50 bp telomere until after a few additional rounds of cell divisions. Together with our earlier data suggesting that *FinalCut* telomeres enable telomerase elongation for telomeres longer than 30 bp (Fig. 1c), and enable protection from degradation in telomeres longer than 40 bp (Fig. 1d–f), we speculate that 20–30 bp represents a threshold below which the telomere is targeted for degradation rather than elongation. In contrast, telomeres spanning 40–50 bp are above the deterministic threshold for dysfunction and therefore have a slower degradation profile. Overall, this difference in the length threshold for these two fundamental telomere characteristics (~30 bp for telomerase elongation and ~40 bp for protection from degradation) suggests that the underlying respective mechanisms are distinct.

As *FC40* and *FC50* cells continuously divide in the above experiment, telomeres are expected to shorten due to the DNA end replication problem prior to dysfunction. According to the current model of telomere shortening[4], one out of two cells should inherit a shorter telomere by the length of the overhang at each cell division (Fig. 3a). Therefore, our results would be compatible with half of *FC40* cells inheriting a short telomere below a deterministic threshold length at each population doubling (PD). For *FC50*, these events would be delayed by ~1–2 population doublings.

### *FinalCut* combined with single-molecule sequencing recapitulates the current model of the DNA end replication problem

To test the idea that *FinalCut* telomeres undergo gradual shortening prior to dysfunction as proposed above, we next precisely traced their shortening at nucleotide resolution using single-molecule sequencing. The DNA end replication problem has traditionally been studied in telomere populations of varying lengths, where the exact shortening path is difficult to identify. Examining telomere shortening in a context where we can generate telomeres of exact lengths therefore, presents a unique opportunity to directly address the phenomenon.

We first examined when the *FinalCut* telomere acquired G-overhangs following Cas9 cleavage, by determining when they are recognized as authentic telomerase substrates in cells. This

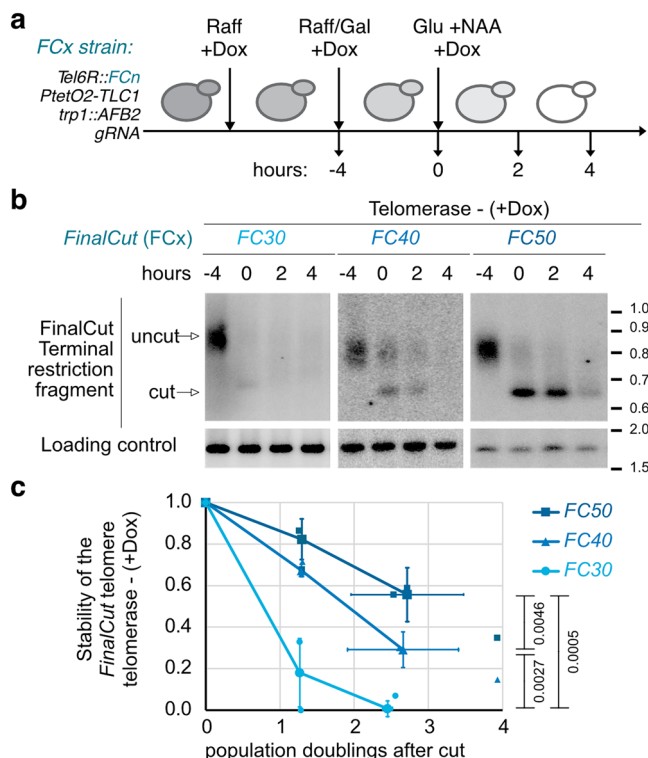

**Fig. 2 | Stability and degradation of the *FinalCut* telomere. a** Experimental design to limit the timing of Cas9 activity and the continuous generation of de novo *FinalCut* telomeres to study their outcome. *FinalCut* strains with indicated genotypes are precultured in Raffinose media containing Doxycycline to turn off the promoter controlling telomerase template RNA gene ($P_{tetO2}$-*TLC1*) throughout the whole experiment. Galactose is then added to induce Cas9 gene ($P_{GAL1}$-*Cas9-AID*) for 4 h ($t = −4$ to 0 h). Cells are subsequently diluted in glucose media containing NAA to degrade Cas9-AID. Samples are taken at indicated time points upon Cas9 degradation. **b** Southern blot of TRF of *Hpa* I digested genomic DNA, prepared at indicated time points from strains with indicated *FinalCut* telomere lengths in the absence of telomerase, probed with the subtelomere 6R and an internal locus in chromosome 11 (Loading control). Molecular weights are indicative for one of the blots. Slightly different migrations/different blots may apply. **c** Quantification of the intensity of the signal of the band representing the "cut" *FinalCut* telomere normalized to the loading control and to the signal measured at $t = 0$ h, as a function of population doublings. Small symbols: values from independent experiments, large symbols: median of three independent experiments, error bars: SD. Statistical differences between *FinalCut* constructs were determined by one-way ANOVA ($p$-value = $8.02 × 10^{-5}$). $p$-values on the right correspond to pairwise comparisons of the slopes of linear regressions (two-sided Student $t$-test). Source data are provided as a Source Data file.

recognition requires the regeneration of a 3′ overhang from the blunt ends created by Cas9 endonucleolytic cleavage[46]. We thus monitored the sequence of *FC50* telomeres over time in the presence of telomerase and developed a dedicated protocol to ensure read depth on individual telomeric sequences and subsequent accuracy (Supplementary Fig. 3a). Inspection of *FC50* sequences at $t = 0$, prior to Cas9 induction, showed that the majority of telomeres contain the *FinalCut* construct and display variable lengths, as expected (Supplementary Fig. 3b). From centromere to chromosome end, they are composed of a common 6R subtelomeric sequence, followed by 50 bp of degenerated $TG_{1-3}$ repeats (*FC50*), the Cas9 target ttDNA, a *Sac* I restriction site, a common stretch of degenerated $TG_{1-3}$ repeats, followed by $TG_{1-3}$ of variable length and divergent sequence among the telomere molecules. This profile is consistent with the expected composition of telomeres in budding yeast. The core region consists of a common $TG_{1-3}$ sequence that is primarily propagated by the semi-conservative

DNA replication machinery. The distal region, however, exhibited sequence divergence among the progeny due to independent, degenerated telomerase-mediated re-elongations of shorter telomeres, as part of the mechanism of telomere length maintenance[40,43].

When cells are incubated in the presence of Cas9 for 4 h and then grown for an additional 6 h in glucose, nearly half of the telomeres displayed sequence divergence starting near the expected Cas9 cleavage site, reflecting de novo telomere re-elongations[43] (Supplementary Fig. 3c). These are likely telomerase-dependent as they are detected in telomerase-positive conditions but not when telomerase is repressed, and fall in the range of telomerase elongations (Supplementary Fig. 3c, inset). We identified position 57 as the most common substrate for telomerase, corresponding to the predicted last nucleotide left by Cas9 double-stranded cleavage (Fig. 3d). However, a substantial number of telomerase-dependent elongations were also detected at position 58. This is presumably due to the variability in Cas9 cleavage, which can create a 1-nt 5′-overhang that is filled by Pol4, as described for some *loci* in various species[47,48]. We concluded that our ttDNA drives staggered cleavage by Cas9, resulting in either a double-stranded blunt end or a 5′-overhang intermediate. Importantly, the data show that all these ends at positions 57 and 58 function as genuine substrates for telomerase, indicating that Cas9-generated ends are likely converted into single-stranded 3′ ends suitable for telomere elongation. Other positions, namely 51 and 46, were also found to share a last common nucleotide, indicating that a telomere shortening event subsequent to Cas9 cleavage must have occurred prior to telomerase elongation. When analyzing the presumed substrates for telomerase elongations, we see that 5′-TCCTGGG-3′ is the most common substrate sequence used by telomerase presumably because it is the most frequent extremity available right after Cas9 cleavage (position 57). Accordingly, annealing of this substrate to the 3′-CCC-5′ of the 3′-ACACACACCCACACCAC-5′ template RNA would be followed by the synthesis of 5′-TGTG-3′, which is exactly what we see (Supplementary Fig. 3c). However, when it comes to shortened substrates, annealing of telomerase RNA template to 5′-GTG−3′ is the most common situation[49]. In this context, the substrates for telomerase elongation at positions 51 and 45 indicate possible shortening steps of ~6 bp from the original *FinalCut* telomere.

We next examined telomeres in cells lacking telomerase. Sequencing of *FC50* and *FC70* over time shows gradual shortening of telomeres at an average ($± SD$) of $4.32 ± 1.06$ and $3.46 ± 0.68$ bp/PD, respectively. These rates are not statistically different ($p$-value = 0.301), and are compatible with previous estimations[29] (Fig. 3e). This suggests that the procedure to generate *FinalCut* telomeres does not affect their subsequent behavior in the cell as compared to native telomeres. Examination of sequence lengths at the individual telomere level revealed a non-uniform distribution (Fig. 3e, f and Supplementary Fig. 3d–g). For *FC50*, at 0 h, corresponding to -1.5 PD since the beginning of galactose induction of Cas9, 51.9% +/−4.7 of telomeres show expected lengths upon Cas9 staggered cleavage at positions 57 and 58, as above. Strikingly, nearly 23.8% +/−2.2 of *FinalCut* telomeres are shortened by 6 bp and end in position 51. At 2 h corresponding to ~2.7 PD, the fraction of telomeres ending at positions 57 and 58 decreased by half, and the ones ending in position 51 is maintained at 30.9% +/−1.8 while a fraction of telomeres ending near positions 40 to 45 starts to emerge. At 4 h, corresponding to 4 PDs, the fraction of telomeres ending at position 51 is now halved to 16.8% +/− 1.5. In sum, our results are compatible with telomeres ending at positions 57−58 being consecutively converted into telomeres ending at position 51, and subsequently into telomeres ending near positions 40−45. These positions correspond to positions re-elongated by telomerase (Fig. 3d and text above). We observed similar results with the longer *FC70*, except that the positions were shifted by approximately 20 bp (Supplementary Fig. 3g). Taken altogether, our data is compatible with a model where *FC50*

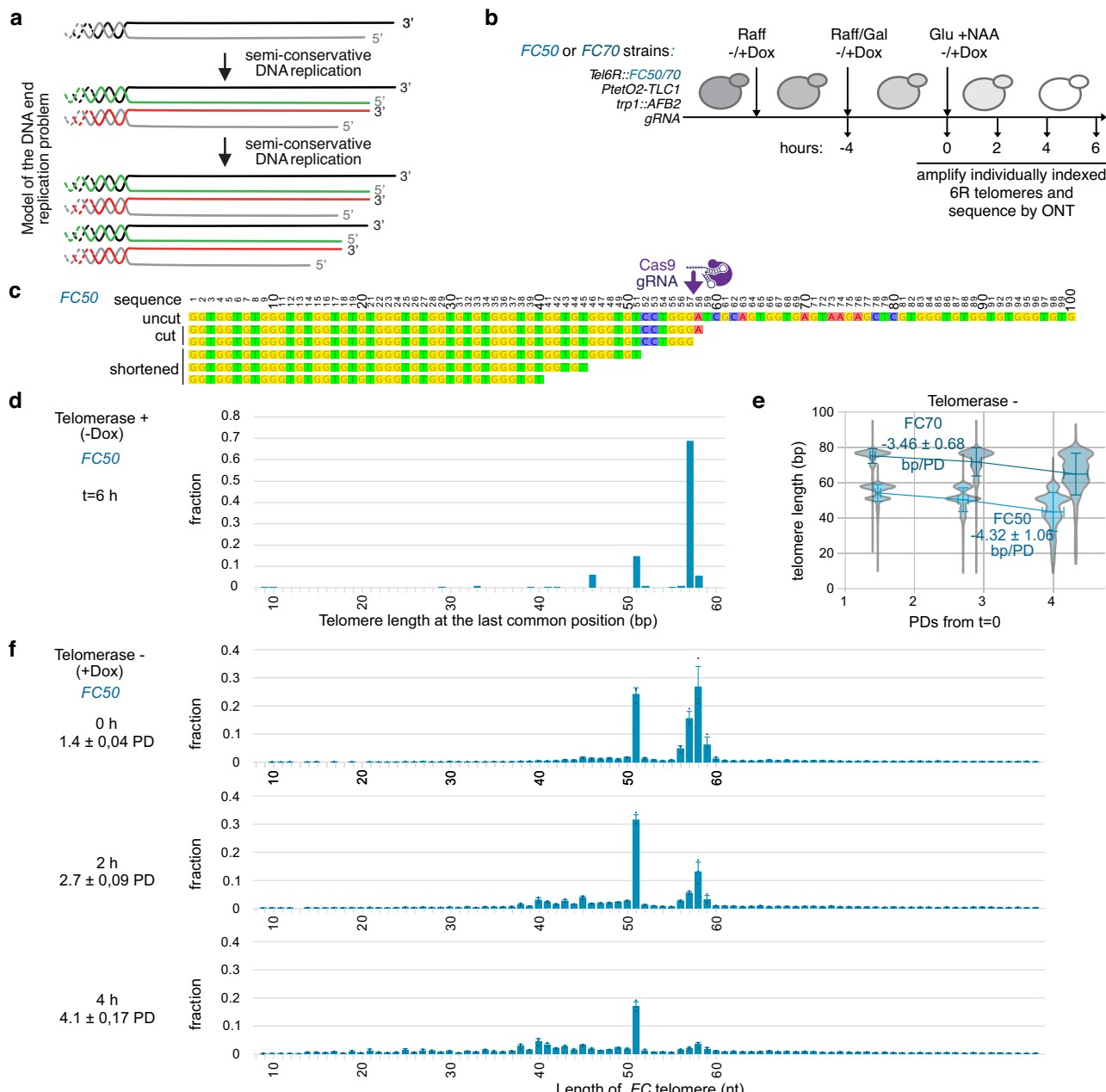

**Fig. 3 | Sequence analysis of individual *FC50* telomeres delineates the DNA end replication problem, revealing a consistent shortening of 6 bp in one out of two telomeres. a** Current model of the DNA end replication problem[4]. As telomeres end in a 3′-overhang, each passage of the replisome in the absence of telomerase results in telomere shortening of one telomere out of two by the length of the overhang, as measured by the 3′-containing strand. Strand synthesized by the leading machinery in red, and the lagging machinery in green. **b** Cells harboring a FC50 or FC70 construct are incubated in Raffinose-containing media, in which doxycycline is added to shut off telomerase or not added. Galactose is then added to induce a degron-tagged version of Cas9 endonuclease (Cas9-AID), before being transferred into glucose and NAA to inactivate Cas9 ($t = 0$ h). With this protocol, short telomeres formed by Cas9 can be tracked and the kinetics of their shortening and/or re-elongations can be assessed by individual telomere sequencing. **c** FC50 primary sequence in uncut, mostly represented cut and shortened states. Relevant positions are indicated. **d** Distribution of *FinalCut* telomeres' last positions presumed to serve as substrates for telomerase elongations obtained at $t = 6$ h of the experiment (**b**) using *FC50* strain in the absence of doxycycline (telomerase +). $N = 227$ from two pooled independent experiments. **e** Changes in lengths of individual indicated *FinalCut* telomeres over population doublings after Cas9 deactivation, without telomerase (+ Dox). Data correspond to the average of 3 pooled independent experiment (FC50: $N = 2769, 4595, 1538$ for $t = 0$ h; $N = 2589, 8840, 2954$ for $t = 2$ h; $N = 4838, 6548, 7594$ for $t = 4$ h; FC70: $N = 1119, 1109, 764$ for $t = 0$ h; $N = 819, 831, 1416$ for $t = 2$ h; $N = 1102, 1018, 745$ for $t = 4$ h. Bars correspond to SD of pools. **f** Magnified view of (**e**), showing the distribution of individual telomere lengths at nucleotide resolution, over time and population doublings after Cas9 deactivation in the *FC50* strain, without telomerase (+ Dox). Dots represent the average obtained in each of three independent experiments, with error bars indicating the standard deviation (SD) of the averages. Source data are provided as a Source Data file.

shortens at a rate of ~6–11 bp on half of telomeres per cell division. This indicates that the overhang length would be ~6–11 bp, which would accommodate an average shortening of ~3–5 bp/cell division measured over a population of telomeres of heterogeneous lengths.

In conclusion, sequencing of *FinalCut* telomeres allows for the detection of telomerase-mediated telomere re-elongations at Cas9-derived cleavages, demonstrating the conversion of blunt ends into canonical 3′-overhangs characteristic of chromosome ends. We further recapitulate the DNA end replication problem with unprecedent

precision in its finest detail by combining the *FinalCut* system with individual telomere sequencing. Therefore, in the absence of telomerase, the shortest telomere in cells undergo gradual telomere shortening in accordance to the current model of the DNA end replication problem, prior to their switch into dysfunction and degradation when reaching 30–40 bp.

## The length of the shortest telomere in the cell limits the proliferation capacity in the absence of telomerase

Our data using *FinalCut* showed that a single telomere of fixed length shortens over time up to a threshold length between 30 and 40 bp, below which it transitions from functional to dysfunctional telomere. However, the precise moment when cells activate the DNA damage checkpoint and enter replicative senescence in relation to telomere status is unclear. We thought to elucidate this event using our experimental system. Since each cell contains a single critically short telomere, this complexity may manifest at the single-cell level, and we therefore analyzed cell fates in the context of these evolving telomere states in single cells, in addition to bulk populations.

At a DSB, the DNA damage checkpoint signaling cascade involves the recruitment of the Mec1/ATR kinase along with its interacting partner Ddc2/ATRIP, where they bind to newly formed ssDNA[50,51]. This complex phosphorylates Rad9 (functionally related to 53BP1 in mammals), which then binds the fork-head associated domains of the Rad53 and Chk1 kinase. Once bound to Rad9, Rad53 is phosphorylated by Mec1, which initiates auto-hyper-phosphorylation of Rad53, and which is required to fully activate and amplify the checkpoint signal, resulting in a cell cycle arrest in G2/M. As a first approach to determine the length of the *FinalCut* telomere activating the DNA damage checkpoint, we assessed Rad53 phosphorylation in bulk *FinalCut* cell populations upon continuous Cas9 induction as described in Fig. 1. We observed that the DNA damage checkpoint was fully activated 3 h after the generation of a *FC20* using *FinalCut*, in accordance with observed rapid degradation of this chromosome end and arm (Supplementary Fig. 4a). This is consistent with *FC20* being recognized as a non-repairable DSB in the cell. In contrast, the induction of the *noFC* control did not alter Rad53 phosphorylation status, as predicted for 1–2 days in culture without telomerase[52,53]. Generation of *FC40*, however, triggered an increased fraction of phosphorylated Rad53 starting at 6 h after Cas9 induction. The extent of Rad53 phosphorylation paralleled the fraction of degraded chromosome ends (Fig. 1f). Therefore, at a population level, both telomere degradation and the activation of the DNA damage checkpoint is gradual over time for *FC40*.

We next turned to experiments assessing DNA damage checkpoint activation at the level of single cells using microfluidics coupled to live-cell imaging, which allowed us to monitor consecutive divisions of single yeast cells starting from onset of telomerase inactivation to replicative senescence onset and cell death (Fig. 4a)[34,54]. Using this experimental setting, we had previously established that part of the telomerase-negative cells undergo a highly variable number of normal cell divisions before undergoing an abrupt and irreversible transition signaled by 2–4 extremely prolonged cell cycles followed by cell death, a route to senescence dubbed in our previous study as *Type A* senescence. We also had found that, shortly after telomerase inactivation, 40–60% of cell lineages undergo, frequent and reversible DNA damage checkpoint cell cycle arrests prior to the terminal 2–4 extremely prolonged cell cycles and death, a separate route to senescence we called *Type B* senescence (Fig. 4b)[34,54]. While consecutive, prolonged, checkpoint-dependent cell cycles have been linked to mutagenic adaptation to DNA damage—a process allowing budding yeast cells to bypass checkpoint-induced cell cycle arrest—type B cells may resume normal cycles after a reversible arrest[34,54]. Assessing replicative senescence at the single-cell level in a time-resolved manner has therefore uncovered layers of intercellular heterogeneity that were not detectable in bulk population assays[24].

We thus introduced the *FinalCut* strains into our custom microfluidics device. Induction of Cas9 generates at *TEL6R* the *FinalCut* telomere that is always the shortest telomere in the cell for all *FC* strains. Cells were allowed to invade the microfluidics circuit microcavity for 16 h in a media containing raffinose as a carbon source, prior to telomerase inactivation via doxycycline. Following 6 h induction of Cas9 expression by galactose addition to the media, the carbon source was switched back to raffinose. Cells were monitored by microscopy throughout the experiment, from telomerase inactivation up to cell death (Fig. 4a). Control cells with active telomerase (*noFC*, noCas9 noDox, Supplementary Fig. 4b) expanded in population without limit with an average of cell cycle duration of ~145 min in raffinose and ~140 min in galactose. In this context, only 4.3% (raffinose) and 4.2% (galactose) of cell cycles were considered "long", i.e above 290 or 224 min for growth in raffinose or galactose, respectively (see "Methods"), and the rate of cell death was 0.5%.

Analysis of the *noFC* control strain grown along the same protocol in the presence of doxycycline to repress telomerase indicated that cells entered senescence with similar dynamics as when they were grown in glucose media, even though the cell cycle durations are longer due to slower growth in raffinose and galactose media (Supplementary Fig. 4b)[34,54]. The overall median lifespan of ~30 generations we observed in the absence of telomerase was compatible with previous measurements; we also observed both *Type A* and *Type B* lineages consistent with previous measurements[34,54]. These observations indicate a comparable rate of telomere shortening and a consistent mechanism of replicative senescence for cells growing in different carbon sources.

In contrast, cells with shorter *FinalCut* telomeres lose most of the inter-lineage lifespan variation (Fig. 4d). *FC0* cells, for example, divide for about ~1 cell division after Cas9 induction before undergoing 1–2 very prolonged cell cycles and dying (Supplementary Fig. 4c), suggesting that loss of a single telomere and exposure of a non-telomeric DNA end causes immediate DNA damage checkpoint activation[55]. A similar profile was found when *FC20* and *FC30* telomeres were analyzed (Fig. 4d), consistent with the immediate degradation that we observed after Cas9 cleavage to generate the *FinalCut* telomere (Figs. 1d–f and 2b, c). Cells with longer *FinalCut* telomeres *FC40*, *FC50*, and *FC70* exhibited a median of approximately 2.5, 4, and 9 cell divisions, respectively, before the onset of very prolonged cell cycles and cell death (Fig. 4d). Interestingly, we also observed that these longer *FinalCut* telomeres resulted in relatively greater inter-lineage lifespan variation, mirroring the telomere length heterogeneity acquired in each cell division due to the DNA end replication problem (Fig. 3a). Taken together, these data provide compelling evidence that the sudden transition from normal to prolonged cell cycles correlates with the length of the shortest telomere in a telomerase-negative cell and the time to reach a telomere length that triggers dysfunction. This is consistent with a model whereby the length of the shortest telomere in the cell determines the onset of senescence. However, because of the growing variability in telomere length and senescence phenotypes over time, establishing a cause-and-effect relationship is not trivial.

## A deterministic threshold model for telomere transition into dysfunction and activation of the DNA damage checkpoint

To complement our cell-based experiments and to more directly test the hypothesis that a sudden transition into dysfunction at a precise threshold telomere length triggers senescence, we turned to mathematical modeling. We previously developed a mathematical model of replicative senescence based on the idea that the length of the shortest telomere in cells drives senescence[30]. Our model was calibrated using quantitative data obtained in microfluidics experiments of cells depleted of telomerase (using an identical repressible $P_{tetO2}$-*TLC1* locus) grown in glucose-based media in the presence of doxycycline, and took into account both *Type A* and *Type B* routes to senescence

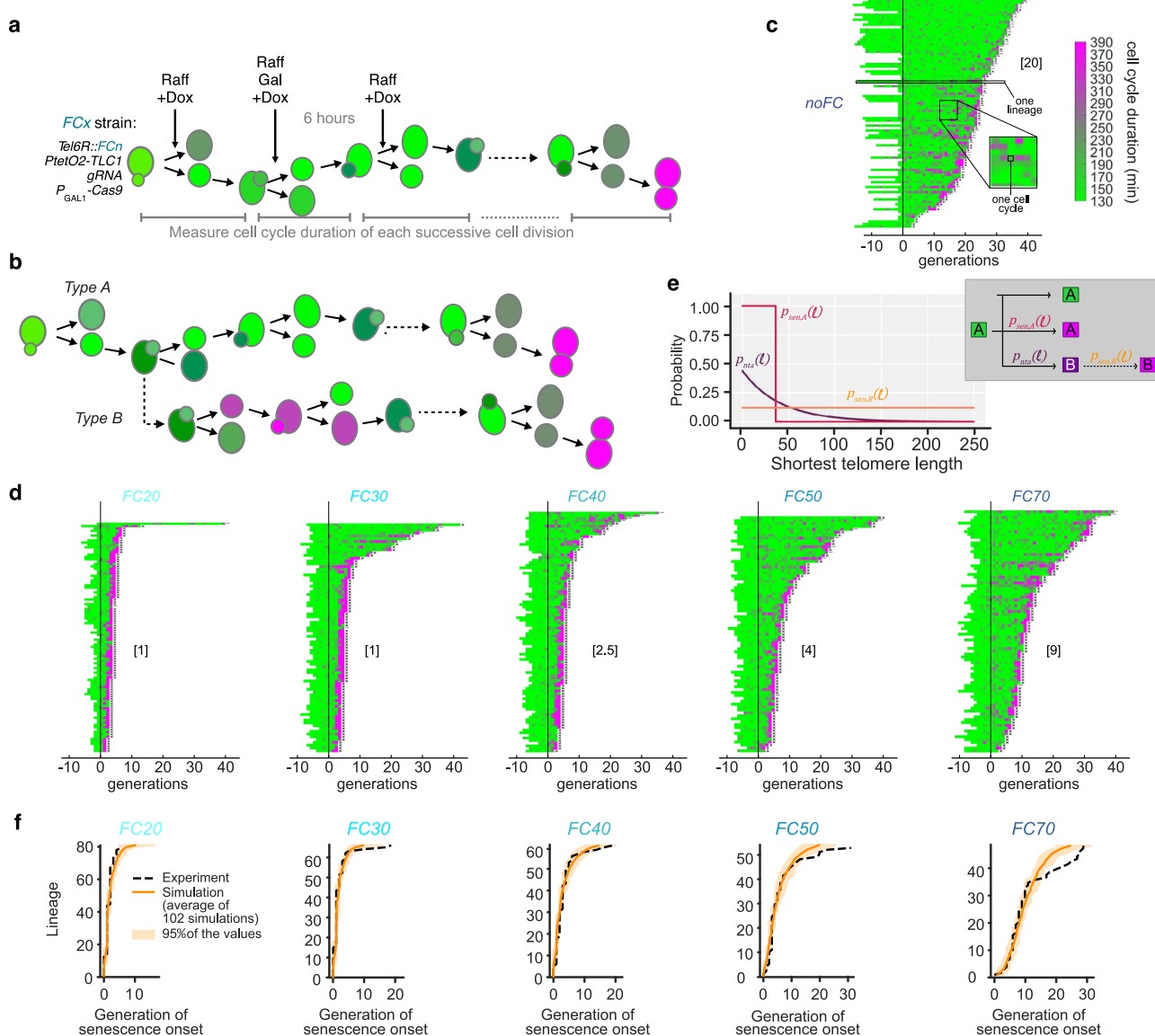

**Fig. 4 | A threshold limit below 40 bp marks the transition from functional to dysfunctional telomere in the absence of telomerase. a** Microfluidics experimental setting. *FinalCut* strains are introduced in a microfluidics circuit. Cells are allowed to invade microcavities through cell divisions. Carbon source is then changed to raffinose and doxycycline is added to inactivate telomerase up to the end the experiment. Galactose is next added for 6 h to induce Cas9. Durations between two cell divisions are recorded, for consecutive cell divisions up to cell death. Variations in cell cycle durations are illustrated by the colour gradient, ranging from green (short) to magenta (long). **b** Description of the two routes to senescence found in microfluidics experiments of control telomerase-negative cells[34]. *Type A* cells undergo senescence in a single abrupt transition from normal cell cycles to terminally prolonged cell cycles (colours as in (**a**)). *Type B* cells undergo series of prolonged cell cycles, followed by normal cell cycles, before ending their route by terminally abnormally long cell cycles. **c**, **d** Display of consecutive cell cycle durations of indicated *FinalCut* cell lineages. Generation 0 corresponds to the addition of Galactose. Each horizontal line corresponds to an individual cell lineage, and each segment to a cell cycle. Cell cycle duration (in minutes) is indicated by the colour bar. X at the end of the lineage indicates cell death, whereas "…" indicates that the cell was alive at the end of the experiment. Data shown corresponds to the sum of at least two independent experiments. Note:

recording was stopped at 5 days post injection into the microfluidics circuit (many lineages have not died at the end of the NoFC experiment (**c**)). In brackets, median generations to senescence onset prior to final senescence, calculated over the lineages actually senescing during the experiment, used in (**f**). **e** Mathematical model of telomerase-negative individual cells recapitulating *Type A* and *Type B* trajectories. Inset: Simplified tree diagram of *Type A* and *Type B* routes shown in (**b**). Colours reflect cell cycle durations as in b. The graph shows the probability of senescence of *Type A* cells as function of the length of the shortest telomere ($p_{sen,A}(\ell)$) (dark red); the probability of *Type A* cells to become *Type B* - when undergoing a first non-terminal prolonged cell cycle - as function of the length of the shortest telomere ($p_{nta}(\ell)$) (dark magenta); the probability of senescence of *Type B* cells as function of the length of the shortest telomere ($p_{sen,B}(\ell)$) (orange). Parameters listed in Supplementary Table 8. **f** Lifespan of individual cell lineages actually entering senescence upon telomerase inactivation, as assessed by microfluidics in this figure (**d**) (dashed line), and 100 simulations of the same number of individual cell lineages with the mathematical model set for indicated *FinalCut* conditions, ordered by lifespan (orange line: average; shaded orange area: range encompassing 95% of the simulated values). Source data are provided as a Source Data file.

(Fig. 4b). The model predicted that the onset of replicative senescence of *Type A* cells is mainly driven by a deterministic threshold length of the shortest telomere in telomerase-negative cells. In contrast, it predicted that the likelihood of transition to the *Type B* lineage gradually increases as the shortest telomere shortens, but that the subsequent onset of replicative senescence in these lineages occurs uniformly randomly (Fig. 4e)[30].

We set our mathematical model to accommodate the experimental conditions of the *FinalCut* system. The parameters concerning cell cycle durations were adapted for cell growth in raffinose and galactose-containing media. The telomere lengths were also adapted to account for the generation of a single telomere set at a given length, according to the kinetics of Cas9 cleavage, as in Fig. 1e. In these conditions, our simulations confirm that in nearly all cells, the *FinalCut* telomere is actually the shortest telomere (Supplementary Fig. 5a and Supplementary Table 7). We next simulated cell growth in the microfluidics device and scanned a range of threshold lengths that would trigger the activation of the DNA damage checkpoint at a precise telomere length for both *Type A* and *Type B* cells. We then compared the simulations to our experimental results. We found a best fit between the experimental data and the simulations for a threshold telomere length of 33–39 bp, which defines the law of probability of transition of *Type A* cells into senescence ($p_{sen,A}$) (Fig. 4f, Supplementary Fig. 5b, c, and Supplementary Table 9). For *Type B* cells, parameters were kept unchanged compared to the general model[30]. *FC70* contained some lineages growing longer than expected (Fig. 4d). While we do not know their exact origin, we suspect that they may represent precursors of post-senescence survivors since their emergence is not included in the mathematical model[30] (see "Discussion").

The close corroboration between our experimental results with the predictions of the mathematical model provide validation for the model's accuracy and assumptions to effectively capture the underlying dynamics of replicative senescence. Taken altogether, our results provide strong in silico and in vivo support that a threshold length exists to activate the DNA damage checkpoint and shift cells suddenly to the last prolonged cell cycles of *Type A* cells (accounting for most cells in the population) and subsequent cell death.

### Telomeres of 30–40 bp are also detected in a context independent of an experimentally induced critically short telomere

Taken altogether, our experimental and mathematical modeling data obtained with the *FinalCut* system show a threshold length below 40 bp at which cell division halts. This is significantly shorter than previous approximations, which estimated a longer telomere length limit of about 70–75 bp from genome sequencing data[14,56]. These latter estimations were derived from sequencing data of telomerase-negative cell populations. We thus wished to test whether telomere threshold lengths we established for telomere protection, clearly below 70–75 bp, could be detected in native telomeres not modulated by *FinalCut*. Of course, cells carrying a critically short telomere are expected to die and be immediately outgrown by proliferating cells, so that the critically short telomere in these cells becomes highly diluted within a population (by 1/32, and in an arrested cell mixed with growing ones) and thus challenging to detect. To circumvent this challenge, we sought to enrich a cell population with short telomeres without compromising cell growth by using the DNA damage checkpoint gene *RNR3* promoter[57] to control the telomerase activity through regulation of the telomerase RNA component, TLC1. We reasoned that in this strain, $P_{RNR3}$-*TLC1*, telomerase activity is restricted enough that cells should perpetually be on the verge of senescence (Fig. 5a). Because telomerase adds degenerated telomere repeats in budding yeast, telomerase-dependent additions can be identified by comparing the sequence of a given telomere from different cells in a population and identifying the divergence point where telomerase added de novo

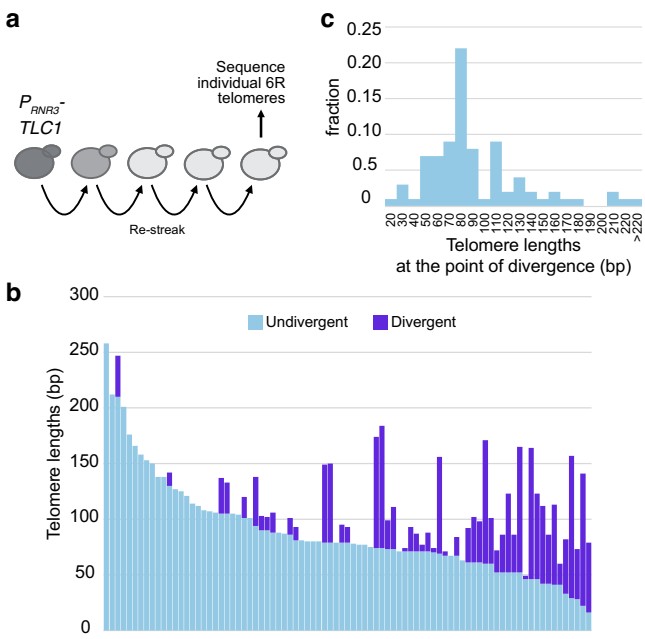

**Fig. 5 | Short telomeres down to 30 bp can be detected in viable cells. a** Experimental setting. A strain expressing TLC1 telomerase template RNA under the control of the promoter of *RNR3*, a downstream effector of the DNA damage checkpoint, was grown for multiple generations. The 6R telomeres were then amplified by PCR, cloned, and sequenced. Resulting degenerated TG$_{1-3}$ telomeric sequences were aligned to identify the point of divergence among the sequences. **b** Representation of the sequence of 6R individual telomeres in which the light blue bars represent the length of common, non-divergent telomeric repeats among *TEL6R* sequences, and the dark blue bars, the length of unrelated, divergent telomeric sequences, indicating de novo telomere elongations by telomerase expressed under the promoter of a checkpoint gene. **c** Distribution of the positions of the point of divergence among the telomeric repeats, suggesting the lengths the 6R telomere reached at the time of the activation of the DNA damage checkpoint. Source data are provided as a Source Data file.

telomeric sequences[43,58] as above (see Supplementary Fig. 3c for an example). In the $P_{RNR3}$-*TLC1* strain, we expect this divergence point to reflect the minimum length of the telomere immediately before telomerase induction for extension, and therefore to reflect the telomere length reached by a given chromosome immediately before senescence regardless of whether or not it is the shortest telomere in the cell.

Consistent with our expectations that $P_{RNR3}$-*TLC1* cells would perpetually be on the verge of senescence, we found that these cells grew poorly, forming heterogeneous-sized colonies on agar plates, which suggested a recurrent activation of the DNA damage checkpoint. Sequencing of telomeres of these cells revealed that they can shorten to a length that activates the DNA damage response, yet is still extendable by telomerase. We observed a wide distribution of telomere lengths with a peak of ~80 bp (Fig. 5b, c). Notably, telomeres below 70 bp accounted for 27% of sequenced telomeres. These data show that fully native telomeres can reach lengths shorter than 70–75 bp without triggering cell death. This is compatible with our conclusion that the lower limit of telomere signaling cell cycle arrest and senescence is below 40 bp.

### Mutation rate increases *in cis* to the shortest telomere in telomerase-negative cells

Our data support a model where a functional telomere gradually shortens due to the DNA end replication problem before transitioning

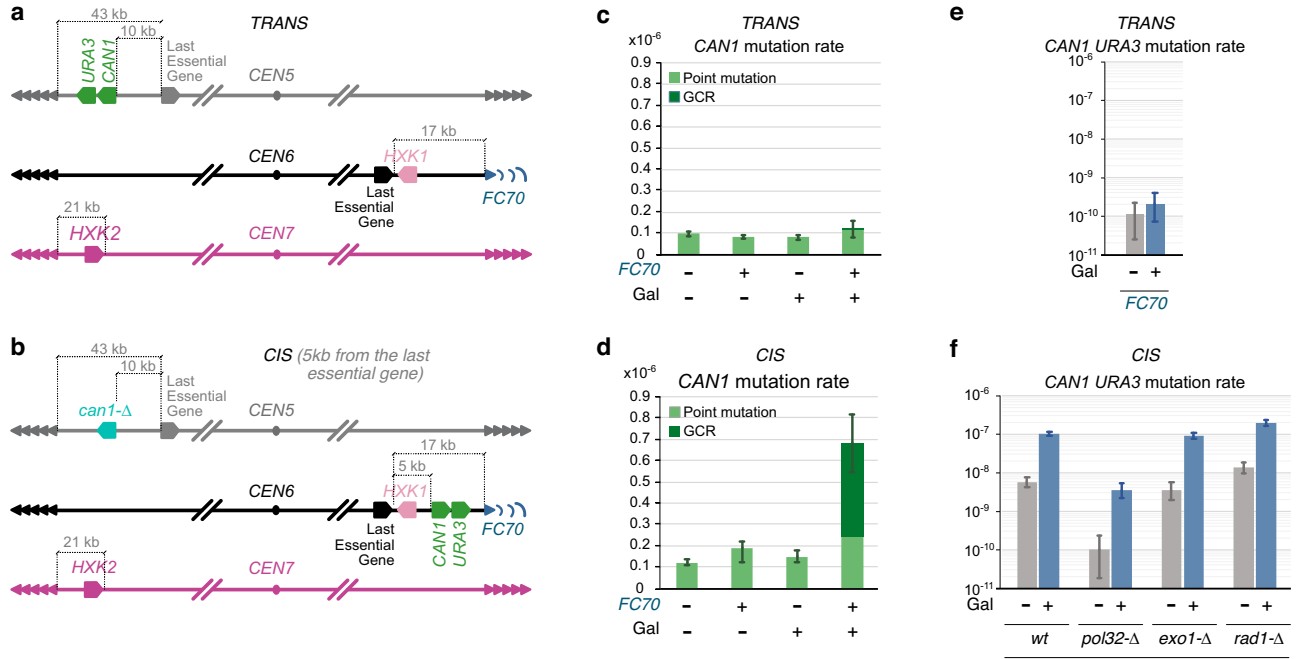

**Fig. 6 | Genome instability in replicative senescence stems from the first telomere reaching a critical short length. a, b** Relevant genomic features of main strains used to quantify genomic instability in *trans* (**a**: yT1690) or in *cis* (**b**: yT1760) of *FC70* telomere by fluctuation assays. Last essential gene of chromosome 5 left arm is *PCM1/YEL058W*; last essential gene of chromosome 6 right arm is *RPN12/YFR052W*. Not to scale. **c, d** *CAN1* mutation rate in *trans* (**c**: strains yT1690 "FC70 +" and yT1694 "FC70 −", noFC) or in *cis* (**d**: strains yT1760 "FC70 +" and yT1759 "FC70 −", noFC)

of *FC70* telomere, according to protocol depicted in Supplementary Fig. 6c. Error bars represent 95% confidence intervals. See Supplementary Table 10 for statistical details. **e, f** Simultaneous *CAN1-URA3* mutation rate in trans (**e**: strain yT1690) or in *cis* (**f**: strains yT1760, yT1870, yT1869, yT1871) of *FC70* telomere, according to protocol depicted in Supplementary Fig. 6g. Error bars represent 95% confidence intervals. See Supplementary Table 11 for statistical details. Source data are provided as a Source Data file.

to a dysfunctional state characterized by degradation along the chromosome arm (Figs. 1 and 2 and Supplementary Figs. 1 and 2). During this process, rare genomic instability can arise, a conserved phenomenon between mammals and yeast that prompts oncogenesis in mammalian contexts[18,19,59]. In budding yeast, telomere-proximal regions have been shown to be involved in gross chromosomal rearrangements in telomerase-negative senescent populations at a frequency of ~10⁻⁸–10⁻⁴, depending on the assay[54,60,61]. We therefore asked whether the shortest telomere had a greater chance of being involved in this phenomenon.

To assess this, we mapped and quantified the genomic instability following telomerase inactivation with respect to the genomic location of a *FinalCut* telomere displaying a known length. We use fluctuation assay as a proxy for genetic instability, and hypothesized that if the telomere shortening causes genetic instability broadly across the genome, we would observe genetic instability in *trans* to the shortest telomere induced by *FinalCut*. In contrast, if genetic instability is more directly connected to dysfunction at the shortest telomere, we would expect to see instability *in cis* to the *FinalCut* telomere on the same chromosome.

We generated strains with a classical reporter cassette containing *CAN1* and *URA3* genes, which allow positive selection of mutants that arise within a population from genetic changes using canavanine- or 5-fluoroorotic acid (5-FOA)-containing media. This cassette was inserted at multiple genomic locations in *cis* to the subtelomeric region adjacent to the *FinalCut* telomere, or in *trans* at another subtelomere (Supplementary Table 9, Fig. 6a, b, and Supplementary Fig. 6). The genomic locations were non-essential in haploid cells as they were positioned downstream of the last essential gene on each respective chromosome arm. When the cassette was inserted in *cis* to the *FinalCut* telomere, we generated two additional sets of strains to evaluate different scenarios: one with a homologous region for *CAN1* present on

another chromosome (where the endogenous *can1-100* is present in chromosome 5, hereafter referred to as the "donor sequence", Supplementary Fig. 6a, b) and one where this region was absent (*can1-*, Fig. 6b).

Mutation rates per cell division were determined by scoring canavanine-resistant (Can^R) colonies and applying fluctuation analysis. These mutants were sampled from multiple independent cultures grown for ~10 population doublings in doxycycline-containing galactose media, and then plated onto canavanine-containing media without doxycycline to re-express telomerase to ensure their survival (Supplementary Fig. 6c). We chose a strain with *FC70* because it allowed gradual telomere shortening for 24 h in the absence of telomerase prior to *FC70* dysfunction (~10 generations, see Fig. 4d and Supplementary Fig. 6d). As negative controls, we used strains without the *FinalCut* construct (Supplementary Tables 1 and 9). These strains showed no increase in mutation rate throughout the experiments, as 24 h of growth in the absence of telomerase was not sufficient to cause an increase in the *CAN1* mutation rate (Fig. 6c, d, Supplementary 6e, f, and Supplementary Table 10)[54,60].

We found that when the reporter cassette *CAN1-URA3* was located in *trans* near *TEL5L* (at 10 kb from the last essential gene of chromosome 5 L arm), the *CAN1* mutation rate remained unchanged 24 h after the generation of the *FC70* telomere (Fig. 6c). In contrast, when the *CAN1-URA3* cassette was placed in *cis* near *TEL6R* bearing the *FC70* construct (either 12.5 kb or 5 kb from the last essential gene on the 6R arm, Fig. 6b and Supplementary Fig. 6a, b), we observed a significant increase in the *CAN1* mutation rate 24 h after the cut, regardless of the presence or absence of a donor sequence (Fig. 6d and Supplementary Fig. 6e, f). These results show that increase in mutation rate during replicative senescence occurs primarily in *cis*, in the vicinity of the *FC70* telomere.

## Genomic instability is localized to the *cis* subtelomere adjacent to the shortest telomere in telomerase-negative cells

We next investigated whether mutations associated with critically short telomeres exhibited a distinctive pattern or mutational signature. To accomplish this, canavanine-resistant (Can^R) colonies from the preceding experiments, selected from multiple independent cultures, were replica-plated on media lacking uracil and on media containing 5-FOA. This allowed us to distinguish gross chromosomal rearrangements (GCRs) from point mutations. We reasoned that [Can^R 5-FOA^R ura^-] colonies likely correspond to GCR whereas [Can^R 5-FOA^S ura^+] colonies likely correspond to point mutations[62]. This analysis revealed that the mutations leading to Can^R colonies in the absence of a critically short telomere in cells and of a donor sequence, both in the *noFC* strains and prior to the generation of *FC70* (0 h in galactose), predominantly corresponded to point mutations. In contrast, the increase in the *CAN1* mutation rate proximal to the *FinalCut* telomere was caused by an increase in GCR (24 h in galactose, Fig. 6c, d).

The basal GCR rate linked to simultaneous loss of *CAN1-URA3* in classical GCR fluctuation assays is described orders of magnitude below the *CAN1* mutation rate in telomerase-negative mutants, prior to senescence onset[63]. Therefore, to better evaluate the increase in GCRs due to *FC70* on GCR levels, we measured the GCR rate by simultaneous plating on canavanine and 5-fluoroorotic acid (5-FOA) (Supplementary Fig. 6g). Our results showed that the *trans* strain loses the 5L chromosome end at a basal rate of ~1 × 10^{-10}, approximately 1000-fold less frequently than the *CAN1* mutation rate—consistent with published data in a similar conditions[63]. Notably, we detected no significant change upon *FinalCut* induction (Fig. 6e and Supplementary Tables 11 and 12). We concluded that a very short telomere in the 6R has no noticeable effect on the stability of another chromosome in *trans*. Yet, we cannot exclude the possibility of effects in *trans*, that would be more difficult to detect, such that telomere-to-telomere fusions and breakage-fusion-bridge cycles[60,64]. In contrast—and consistent with the results from sequential plating on canavanine followed by 5-FOA—we observed an ~18-fold increase in GCRs in *cis* to *FC70* (Fig. 6f and Supplementary Tables 11 and 12). Taken together, these findings demonstrate that the shortest telomere in a telomerase-negative cell is the most prone to initiating genomic instability, which most frequently involves the adjacent chromosome arm.

## Non-reciprocal translocations at subtelomere 6R serve as a signature of genomic instability at the shortest telomere in the absence of telomerase

In the presence of donor sequence, a substantial portion of Can^R colonies corresponded to GCRs in all conditions (Supplementary Fig. 6e, f). To gain insight into their karyotype, we separated yeast chromosomes by pulsed-field gel electrophoresis (PFGE) of [Can^R 5-FOA^R ura^-] colonies obtained 24 h after the generation of the *FC70* when the cassette was located 12.5 kb from the last essential gene (colonies obtained with strain yT1696). As controls, we established the karyotype of the initial strain [Can^S 5-FOA^S ura^+] and of a [Can^R 5-FOA^S ura^+] colony obtained at 0 h of galactose. For the [Can^R 5-FOA^R ura^-] colonies, we detected a specific translocation involving chromosome 6, when compared to the [Can^S 5-FOA^S ura^+] and [Can^R 5-FOA^S ura^+] karyotypes (Supplementary Fig. 7a). Subsequent Oxford Nanopore Technologies (ONT) whole genome sequencing of [Can^R 5-FOA^R ura^-] colonies obtained 24 h after the cut indicated a specific nonreciprocal translocation involving the *CAN1* sequence of the cassette proximal to *FC70* at chromosome 6 and the donor sequence at chromosome 5 (Supplementary Fig. 7b–f). The donor sequence (*can1-100* locus and its flanking regions) shares perfect homology across 2674 bp (ChrV:31457-34131) with the *CAN1-URA3* cassette, except for a single nucleotide causing the ochre mutation (AAA-to-TAA nonsense change at codon 47). We reasoned that the presence of those two highly homologous sequences, both located in non-essential regions, are responsible for this observed specific nonreciprocal translocation. Moreover, this high homology between the donor sequence and the cassette could explain the substantial portion of Can^R colonies corresponding to GCRs in all conditions (Supplementary Fig. 6e, f).

We next analyzed the *cis* strains lacking the donor sequence (Supplementary Tables 1 and 9). Results demonstrated that deletion of the *can1-100* locus and its flanking regions (*can1-Δ*) reduced spontaneous GCRs, although those associated with *FC70* telomere at 24 h in galactose were still observed (Fig. 6d). ONT analysis of three [Can^R 5-FOA^R ura^-] colonies, obtained 24 h upon induction of *FC70* telomere, revealed several nonreciprocal translocations involving the paralogous gene loci *HXK1* (subtelomere 6R) and *HXK2* (subtelomere 7L), which share 78.1% sequence identity (Fig. 7 and Supplementary Fig. 7g, h). Taken altogether, our results demonstrate that the shortest telomere in the cell is an "at-risk" factor that increases genomic instability and promotes GCR. The characteristic of GCR being non-reciprocal translocations results in the re-lengthening of telomeric sequences at *TEL6R* possibly enabling temporary senescence escape.

## GCRs in *cis* of the shortest telomere in telomerase-negative cells depend on Pol32

The non-reciprocal translocations detected in the *cis* GCR assay resemble BIR products. Notably, whole genome sequencing revealed translocation breakpoints at native subtelomeric homology regions. To directly test the BIR mechanism, we deleted *POL32*, which encodes a non-essential subunit of DNA polymerase δ required for BIR[16,65,66]. In strains carrying the *CAN1-URA3* marker in *cis* of FC70—positioned 5 kb from the last essential gene—and *can1-Δ*, *POL32* deletion reduced GCR rates by 29-fold under induced *FC70* conditions and 56-fold in its absence, compared to the *POL32* wild-type context. This suppression of GCRs in *pol32-Δ* strongly supports BIR as the primary mechanism driving GCR formation in the absence of telomerase, when subtelomeric homology regions are present. Taken together, our results support a model in which the absence of telomerase, combined with the presence of homology regions, drives genomic instability through BIR, prior to, or at least independently of, post-senescence survival, which also relies on BIR.

Given that dysfunctional *FinalCut* telomeres undergo substantial degradation (Fig. 1f and Supplementary Fig. 1d, e), we hypothesized that this end degradation could trigger BIR involving internal subtelomeric elements. Accordingly, both Exo1 exonuclease and Rad1 endonuclease have been implicated in increased genomic instability in telomerase-negative populations[61]. However, neither Exo1 nor Rad1 is strictly required for BIR[67,68]. While Exo1 mediates 5′ to 3′ resection of DNA DSBs, generating single-stranded DNA for Rad51 filament formation and strand invasion, deletion of *EXO1* paradoxically increases BIR frequency[69]. Similarly, GCR rates in the classical GCR assay—which lacks homology between the last essential gene and the GCR reporters—are elevated in *exo1-Δ* strains, compared to *EXO1*[70]. On the other hand, Rad1, though involved in processing non-homologous tails during homology-directed repair events, does not play a major role in the core BIR pathway itself[67].

To solve this paradox, we tested the involvement of these nucleases in the GCRs in *cis* to *FC70*. We introduced *exo1-Δ* or *rad1-Δ* mutations into our *cis* GCR assay strain (Fig. 6b). We found that both mutations had only modest effects on GCR rates compared to wild type, with changes typically less than threefold, which is generally considered biologically insignificant in this context[71] (Fig. 6f and Supplementary Tables 9, 11–13). This suggests that Exo1 and Rad1 are not essential for BIR-mediated GCR formation in this context. Alternatively, *EXO1* or *RAD1* mutations do actually decrease BIR-mediated GCR rate in the absence of telomerase, but other repair mechanisms may be upregulated in the absence of appropriate end degradation, resulting in an apparent unchanged GCR rate.

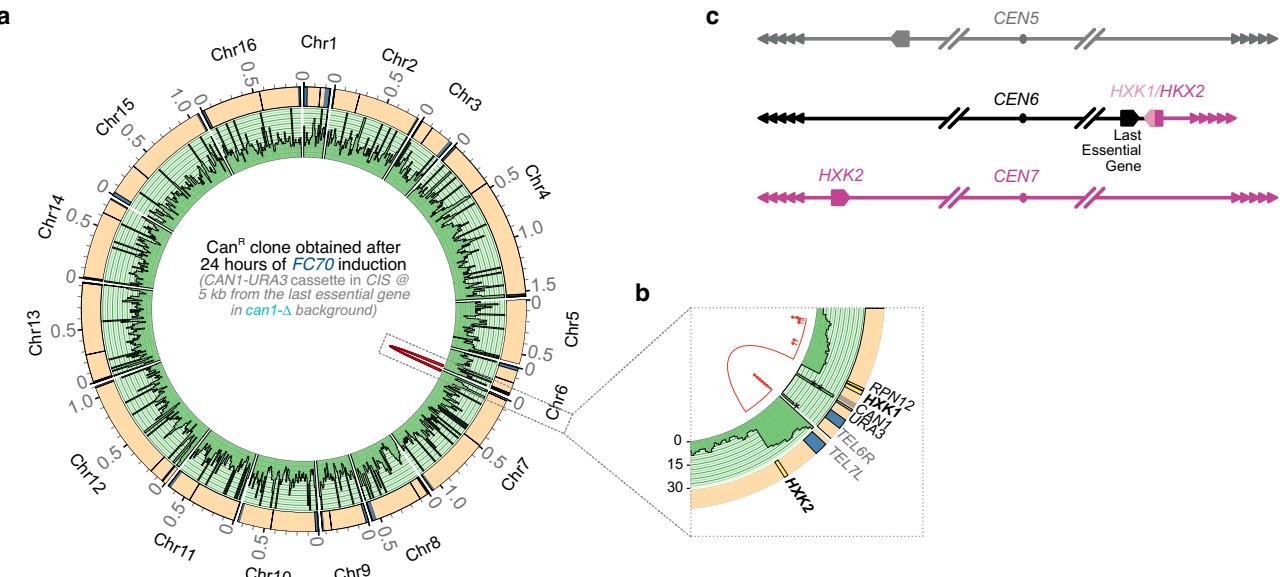

**Fig. 7 | Genomic instability at the shortest telomere in the cell is characterized by non-reciprocal translocations enabling senescence escape. a** CIRCOS plot showing the genome of a representative [Can$^R$ 5-FOA$^R$ ura$^-$] colony classified in this study as GCR, obtained after 24 h of Cas9 induction in the absence of telomerase of cells harboring a *FC70* telomere, the *CAN1-URA3* cassette at 5 kb from the last essential gene on chromosome 6R and in a *can1-Δ* background (initial strain: yT1760, Fig. 7b; experiment Fig. 7e. The outer ring represents reference *S. cerevisiae* chromosomes. Subtelomeric regions are displayed in blue. Centromeres are represented by a black line. Chromosome coordinates are indicated in Mbp. The inner ring displays the read-depth, ranging from 0× to 30×. Extremities of splitted reads are shown as red bridged lines. **b** Zoom of indicated region of (**a**) showing loci involved in the breakpoint region, including *HXK1/2* in bold. The solid red line represents the longest splitted read. Red dots indicate the extremities of splitted reads. (*) Reads containing exclusively TG$_{1-3}$, which map telomeric regions indiscriminately. (**) Reads mapping a region of the *CAN1-URA3* cassette (in grey) containing a short plasmid-derived sequence (60 bp), also present at the *AFB2-TRP1* locus. Initial strain: yT1760. *N* = 3. **c** Proposed model for the chromosomal rearrangement involving chromosomes 6 and 7 of Can$^R$.

## Discussion

Telomere shortening in the absence of telomerase limits cell proliferation by triggering cell cycle arrest and is considered a crucial barrier to oncogenic transformation. However, this same phenomenon promotes genomic instability on the path to cellular senescence. In this work, we use *S. cerevisiae* and an experimental system of telomere length control to decisively demonstrate that the shortest telomere in cells is pivotal to both the onset of replicative senescence and genomic instability. We demonstrate that the transition into dysfunction of the shortest telomere in a cell provides the mechanistic link between these two seemingly opposite but intimately related processes. Our results outline a step-by-step progression from gradual telomere shortening to dysfunction at a critical length. Telomere dysfunction is characterized by extensive inward chromosome degradation from the telomere and causes a DNA damage response checkpoint-induced arrest in cell cycle progression. This telomere degradation enhances the likelihood of presenting subtelomeric elements as substrates for gross chromosomal rearrangements.

Our *FinalCut* system, in which we use genetically controlled Cas9 to generate a single near-native telomere of precise length per cell, offers key advantages to previous approaches to generate and study short telomeres. For instance, in the HO-cut system, the 24 nt non-telomeric HO endonuclease binding site is introduced in a subtelomeric region downstream of the last essential gene[72,73], leaving a non-telomeric terminal end within a truncated subtelomeric region. The FRT or the Cre-LoxP systems, which achieve the production of short telomeres of variable length by excising internal telomeric repeats, generates shortening to a variable length[29,31,74]. In contrast, the *FinalCut* system exposes a defined length of nearly identical telomeric sequences within a native subtelomeric environment, making it the most accurate system for studying telomere dynamics and the impact of telomere shortening to date.

The *FinalCut* system allowed us to observe the DNA end replication problem with unprecedented accuracy, starting with the generation of a near-native telomere of a defined length. We demonstrated that this telomere rapidly converts into a telomerase-extendible 3′-overhang, enabling it to function as a native telomerase substrate. Using *FinalCut* coupled with single-molecule sequencing, we provide a solution to a long-standing controversy regarding the number of shortened telomeres that are needed to drive onset of replicative senescence. Early studies have associated telomere length with cell proliferation capacity in eukaryotes and suggested that short telomere length must limit cell divisions[8,75–78]. Experiments in human primary cells cultured ex vivo have indicated that approximately five short telomeres are required to arrest the cell cycle at the Hayflick limit[79]. However, in vivo, old baboons accumulate dermal fibroblasts displaying a single focus containing both telomeric DNA and DNA damage activation markers[80,81], suggesting that a single telomere could be enough to trigger replicative senescence (although whether this single focus corresponds to the clustering of several telomeres or to a single short telomere remains unclear). These in vivo findings raise the possibility that the observations in the experiments could be the result of strong selection bias for cells that have bypassed the first steps of replicative senescence. In budding yeast, our previous work suggested that an induced short telomere could accelerate replicative senescence and that senescence heterogeneity would be compatible with a single short telomere being sufficient to trigger replicative senescence[25,30,31,34,36]. We now employ single-telomere and single-cell analyses, combined with mathematical modeling, to unequivocally demonstrate that in telomerase-negative budding yeast cells, the first telomere that reaches a deterministic threshold short length between 30 and 40 bp is both necessary and sufficient to trigger the onset of replicative senescence in a large majority of cells. Whether a single short telomere is necessary and sufficient also in other eukaryotes is a possibility that requires further careful testing.

Using single-molecule sequencing, we were able to visualize telomere shortening with nucleotide resolution over approximately two rounds of replication fork passage, and directly demonstrate that the resulting telomere length distributions align with the current model's prediction of asymmetrical telomere shortening, where one telomere out of two shortens (Fig. 3a)[4]. Our findings also show that the length of the 3′ overhang at the *FinalCut* blunt end varies from 6 to -11 nucleotides. According to the current model, the overhang length upon telomere processing results from two parameters related to C-strand synthesis: the positioning of the primase catalytic site within the CST-Polα-primase complex relative to the chromosome end and the length of the RNA primer of the last Okazaki fragment, which is eventually degraded by a yet unknown mechanism. Given that telomere shortening in budding yeast is minimal (~3–5 bp per cell division), these parameters must be tightly regulated. However, the detected variability in shortening steps suggests some flexibility. This flexibility could arise from either the CST-Polα-primase complex positioning the RNA primer's start at various sites or from variability in the RNA primer's length. Recent studies have highlighted the CST complex's role in addressing the DNA end replication problem and stabilizing the G-overhang[5,82]. We believe the technologies developed in this work will soon clarify additional unresolved aspects of the DNA end replication problem.

What happens at the specific threshold length we identified, below which degradation extends inward into the chromosome? One can hypothesize that the telomere structure, composed of DNA, proteins, and RNA, breaks down abruptly at the threshold length. Perhaps the higher-order molecular Velcro, composed of Rap1-Rif1 and Rif2[83,84], can no longer maintain its integrity below a certain telomere length and collapses, preventing the anti-checkpoint function of the telomeres[85]. It is also possible that multimerization of CST, required for telomere elongation, requires a certain amount of DNA to synergize effectively and maintain protection from nucleases in G1 or during the passage of the replication fork[82,86]. Future work to define the telomere composition before and after telomere dysfunction will be greatly facilitated by the *FinalCut* system. It will also be important to define DNA damage checkpoint status in relation to telomere structure. In previous work, we and others have suggested a transition from a Tel1-bound state of the telomere to a Mec1-bound state at replicative senescence onset[31,32,74,87]. We suggest that this transition corresponds to gradual telomere shortening, involving regulated yet consistent Tel1 activation at each replication fork passage, followed by abrupt degradation potentially characterized by transient accumulation of ssDNA capable of activating Mec1. While we observed in these studies the disappearance of the telomeric signal at the point of dysfunction on both strands, further investigation is needed to determine if this indicates differential or delayed in time degradation of one strand compared to the other.

In our experiments, we demonstrated that genome instability upon telomerase inactivation is initiated in the vicinity of the first critically short telomere, the *FinalCut* telomere, while the stability of the rest of the genome does not appear to be compromised. Furthermore, the analysis of individual Can$^R$ colonies that arose from mutations following *FinalCut* in *cis* revealed that the vast majority of mutations associated with the shortest telomere corresponded to GCRs. These findings align with previous investigations observing an increase in genomic instability during replicative senescence in budding yeast, generally proximal to chromosome ends in comparison to the centromere-proximal region[60,61]. We now propose that these previous genomic instabilities may correspond to the subset of cells in the population where the mutation reporter is located adjacent to the shortest telomere, which is different in different cells. This result is also reminiscent of observations made in human MRC5 fibroblast cells, where the relatively short telomere on chromosome 12p is frequently involved in rearrangement events during telomere crisis[21].

Additionally, the consistently short telomere on chromosome 17p has been associated with a higher risk of esophageal cancer in individuals[88–92]. We thus speculate that the mechanisms driving genomic instability during replicative senescence, as demonstrated in yeast, may also account for these correlations found in humans. For instance, the shortest telomere, such as the 17p in humans, would be at the greatest risk of genomic instability because it would also be the first to undergo a risky specific processing, required to trigger replicative senescence.

Whole genome sequencing of Can$^R$ colonies selected upon *FinalCut* induction revealed that the GCRs found in *cis* of *FinalCut* telomeres correspond to nonreciprocal translocations. These translocations are characterized by junctions at homologous regions within subtelomeric repeated elements, such as 5L or 7L, serving as a mutational signature. These observations now enable us to interpret similar events found in conditions where the shortest telomere was random and not identified, mentioned above[60,61], as likely having occurred at the shortest telomeres in the ancestral cells they derived from. Notably, in our case, the subtelomere of the 6R arm is systematically truncated. The most parsimonious hypothesis is thus that genomic instability follows the chromosome degradation we detect upon telomere dysfunction at telomeres below the length threshold. While much will be needed to understand Exo1 and Rad1 hypothetical contributions to the process[61], redundant pathways likely exist to generate single-stranded DNA up to internal subtelomeric regions[50,74]. These pathways could expose sufficient ssDNA for Rad51-dependent strand invasion of a donor chromosome, and create a new 3′ end, thereby enabling Pol32-dependent DNA synthesis, initiating BIR[66,68].

Given that the nonreciprocal translocations detected in this work involve the copy of another subtelomere and telomere, likely by BIR, the resulting cell should deactivate the DNA damage checkpoint. Technically, this cell can potentially divide for some additional generations before the second shortest telomere eventually triggers replicative senescence later on. Therefore, cells having undergone through this type of GCR share the required features that characterize senescence escape, and could therefore constitute post-senescent survivor precursors maintained by ALT via BIR. We thus speculate that the shortest telomere in cells, in addition to being the mechanistic link between proliferation limitation and genomic instability, also serves as a basis for the initial events of ALT. However, much remains to be discovered, as the type of GCRs identified in this study likely represents only a small subset of all possible events, given the selection for the loss of the *CAN1-URA3* cassette at specific genomic locations.

In conclusion, since many of our findings in budding yeast may be conserved in human cells, this work offers a potential paradigm for understanding the link between cellular aging and increased cancer risk, as well as the etiology of syndromes associated with short telomeres.

## Methods

### Yeast strains

Yeast strains used in this work are listed in Supplementary Table 1. All strains used in this work are based on yT1487, the *noFC* control. yT1487 is a derivative of G49 (*RAD5 ADE2* W303 strain obtained from M. Lisby) in which *TLC1* is expressed from the repressible promoter $P_{tetO2}$ by doxycycline[34] and $P_{ADH1}$-*AFB2* gene is inserted at the *trp1* locus by transformation of Ylp204-PADH1-AFB2 (a gift a from Helle Ulrich). *FinalCut* strains were constructed by first creating a Cas9-induced DSB at the junction between the 6R subtelomere and *TEL6R* followed by repair with a donor fragment containing a homologous sequence for the 6R sub-telomere, a stretch of 26 bp-insertion enabling the removal of the previous Cas9 target and the insertion of a *Hind* III restriction site, a variable number of telomeric repeats n(TG$_{1–3}$), the target DNA for Cas9 (ttDNA) and a stretch of telomeric repeats. In practice, yT1487 was transformed simultaneously by pT63 and *Sph* I–*Eco* RI

digested-plasmid pT182, pT177, pT176, pT175, pT174, pT187 or pT188 (Supplementary Table 3). Transformants were selected on media lacking leucine, subcloned on YPD, and checked for [leu⁻]. After loss of the Cas9-expressing plasmid, constructions were checked by Southern blot and regularly by PCR using oT1572-oT1881 right before their use in subsequent experiments. In yT1612 and yT1614, the selectable marker cassette CAN1-URA3 was inserted by transformation at 4 kb or 12 kb from TEL6R (12.5 or 5 kb from the last essential gene on 6R chromosome arm), to generate the strains yT1696, yT1698, yT1710, and yT1712. The CAN1-URA3 cassette was amplified by PCR from the pT196 plasmid using the oligos oT2126-oT2127 or oT2130-oT2131 for the insertion at 4 kb or 12 kb from TEL6R, respectively. Transformants were selected on media lacking uracil and the expression of the CAN1 gene was evaluated by the absence of growth on plates containing canavanine. Insertion of the CAN1-URA3 cassette at 4 kb or 12 kb from TEL6R was confirmed by PCR using oT1894-oT1324 and oT1894-oT2139 primer pairs, respectively. Strains lacking the can1-100 locus were engineered by CRISPR-Cas9 (expressed from pT194), followed by repair with the ssODN oT2276 in yT1612 and yT1614 strains. Subsequently, the CAN1-URA3 cassette was inserted at 12 kb from TEL6R as previously described. POL32, EXO1, and RAD1 were deleted using a PCR amplified fragment of pFA6-natNT2[93] with the primer pairs oT633-oT634, oT586-oT587, and oT2424-oT2425 to transform yT1760 and generate yT1869, yT1870, and yT1871, respectively. A list of primers used in this work is provided in Supplementary Table 4.

## Plasmids

Plasmids are described in Supplementary Table 3. Cas9-expressing pT63 was constructed by cloning annealed oT1450 and oT1451 into Bpl I-digested bRA90, as described[94]. Cas9-expressing pT117 was constructed by NEBuilder® HiFi DNA Assembly of 4 fragments obtained by PCR of bRA77[94] using primer pairs oT1806-oT1667; oT1666-oT1750; oT1742-oT1749 and plasmid pHis−AID*−9myc[45] using oT1741-oT1807. pT122 was constructed by first cloning annealed oT1540-oT1541 into Bpl I-digested bRA77, which provided the PCR amplicons with oT1829-oT1832 and oT1831-oT1834 to assemble with the PCR amplicons of pRS305 with oT1833-oT1836 and oT1830-oT1835 using the NEBuilder® HiFi DNA Assembly. pT126 was constructed by digesting pT117 with Bpl I and ligating to annealed oT1550 and oT1551. Plasmids pT182, pT177, pT176, pT175, pT174, pT187, and pT188 were synthesized by Genecust based on derivatives of pURTel[95] and pT26[39] harboring a cloned fragment of subtelomere 6R followed by cloned duplex oT1713-oT1714 Hind III-Xho I fragment (replacing original URA3 marker) and by TEL6R telomeric repeats. To create the P_{RNR3}-TLC1 gene, the RNR3 promoter was amplified from yeast genomic DNA using primers FW_RNR3pr + HindIII and REV_RNR3pr + BamHI and inserted into plasmid pRS305 digested with HindIII and BamHI. TLC1 was amplified from yeast genomic DNA using primers FW_TLC1 + BamHI and REV_TLC1 + XbaI and inserted into the pRS305-P_{RNR3} plasmid digested with BamHI and XbaI to create plasmid pIK2. Linearized pIK2 can then be inserted into the yeast genome at the leu2 locus. pT194 was constructed by cloning annealed oT2057 and oT2058 into Bpl I digested bRA90. pT196 was constructed by NEBuilder® HiFi DNA Assembly of 3 fragments obtained by PCR of CAN1 locus from a BY4741 background strain with oT2082 and oT2085 and PCRs of pRS306 with primers oT2028 and oT2083 on one hand and oT2084 and oT2029, on the other hand.

## Galactose induced Cas9 cleavage of FinalCut telomeres

FinalCut strains transformed with pT126 with the one-step protocol[96] were inoculated in 10 ml of synthetic media lacking leucine containing 2% raffinose (S -LEU R), grown for 5 h at 30 °C at 220 rpm and diluted to $OD_{600nm}$ = 0.01−0.02 in S -LEU R either with or without doxycyline at 30 µg/ml. Exponentially growing cells were collected 6 h to overnight growth. Cells were sampled for time point 0 h, washed and diluted to $OD_{600nm}$ = 0.2 in synthetic media lacking leucine containing 2%

raffinose and 3% galactose (S -LEU RG) either with or without doxycyline at 30 µg/ml, and grown for 3 h. Sample collection, dilution to $DO_{600nm}$ = 0.3 were repeated each 3 h with 9 h time point being the last one collected. In experiments in which Cas9 is switched off, cells were grown similarly, with the following differences. At 4 h after galactose addition, cells were washed and diluted to $OD_{600nm}$ = 0.1 in YPD either with or without doxycyline at 30 µg/ml and 0.25 mM of NAA. Samples were collected every two hours ($t$ = 4 + 2, 4 + 4, 4 + 6). In microfluidics experiments, FinalCut strains were transformed with integrative pT122 and selected on media lacking leucine. Exponentially growing cells were injected in the microfluidics circuit and allowed to divide in YPD to invade the cavities for 12−24 h before the medium was switched to YP-2% Raffinose either with or without doxycyline at 30 µg/ml. The induction of Cas9 was achieved by switching the medium to YP-2% Raffinose, 2% galactose either with or without doxycyline at 30 µg/ml for 6 h before changing media back to YP 2% Raffinose either with or without doxycyline at 30 µg/ml. For each set of experiments, at least two independents original transformants for the FinalCut constructs were involved in triplicate experiments.

## Southern blots

Genomic DNA was extracted from cultures using a standard phenol: chloroform:isoamyl (25:24:1) purification procedure and isopropanol precipitation. 500 ng of genomic DNA were digested with Hpa I (NEB), ethanol-precipitated, resuspended in gel loading dye (Purple 6×, NEB), and resolved on a 1% agarose gel for 16 h at 50 V. The gel was then soaked in 0.25 M HCl followed by 20 min incubation in 0.4 M NaOH, 1 M NaCl. Transfer was performed overnight by capillarity to a charged nylon membrane (Hybond XL, GE Healthcare or Hybond N+ cytiva #RPN303B). Membranes were pre-hybridized for 2 h at 65 °C in rotating tubes before radioactive probes were added. Random priming reaction by Klenow Fragment (3'→5' exo-, NEB) using α-³²P-dATP was employed to label amplicons of gDNA with oT1959-oT1960 (TRF FinalCut probe), with oT2189-oT2190 (Chromosome arm 6R internal band) or with oT1963-oT1964 (loading control). The telomere-specific oligonucleotide probe oT2032 was 5'-labelled using T4 polynucleotide kinase (NEB) and γ-³²P- ATP. Hybridization was performed overnight at 65 °C. Membranes were washed increasing stringency's buffers (Supplementary Table 5). Membranes were exposed to BAS Storage Phosphor Screens (GE Healthcare) for 2 days and imaged with a Typhoon FLA 9500 scanner (GE Healthcare). Band intensities were measured using ImageJ 1.53a[97]. The smears/bands corresponding to the 6R telomere were first normalized to the loading control band prior to other calculations. Uncut and cut quantities were obtained by a second normalization on the 0 h "uncut" time point. Degraded quantities were estimated subtracting the sum of uncut and cut to the sum of uncut and cut at 0 h. In experiments in which Cas9 is switched off after 4 h in galactose, the stability of the FinalCut telomere at different time points was estimated by normalizing to $t$ = [4 + 0].

## Western blot

Protein analyses were performed as described[53]. In brief, 5 $OD_{600nm}$ of cells were lysed in NaOH 0.2 M for 10 min on ice and proteins precipitated at a final concentration of 0.5% trichloroacetic acid by incubation again on ice for 10 min. After centrifuging for 10 min at maximum speed at 4 °C, the pellet was resuspended in 50 µL of Laemmli 4× buffer and denatured at 95 °C for five minutes. Protein samples were electrophoresed on 7.5% of 37.5:1 Acrylamide:Bis-acrylamide (Sigma-Aldrich #A3699) for Rad53 detection or 4−15% Mini-PROTEAN® TGX™ precast polyacrylamide gels (BioRad) for Cas9 detection. Proteins were then transferred to a nitrocellulose membrane (Amersham Protran 0.45 NC, GE HealthCare), stained with Ponceau red (Sigma-Aldrich). Membranes were blocked with 5% milk and incubated with the appropriate primary antibody: Rad53 (1:1000, Abcam #ab104252), c-Myc (1:2000, Sigma-Aldrich #M5546), FLAG

(1:2000, Sigma-Aldrich, SAB431135), and Pgk1 (1:20,000, Abcam #ab113687). After washes, the primary antibodies were recognized by the horseradish peroxidase-coupled secondary antibody (1:10,000, Sigma-Aldrich #A9169). The signals were revealed using an electrochemiluminescence reagent (ClarityWestern ECL, Biorad) and recorded using the ChemiDoc Imaging System (Biorad).

## TeloPCR
Amplification of 6R-specific telomeres was performed with 40-100 ng of gDNA as described[40]. DNA was denatured in a volume of 6 µl at 95 °C during 5 min and end-tailed in 1× NEB4 buffer (NEB), 100 µM dCTP and 1 unit of Terminal Transferase (NEB #M0315) at 37 °C 30 min, followed by a 10 min incubation at 65 °C and 5 min at 94 °C. PCR mix was added to obtain final concentrations of 200 µM of dNTPs, 0.5 µM of oT883, 0.75 µM of oT2387, 1× Taq ThermoPol Mg2+ free buffer (NEB #B9005), and 2.5 U of Taq polymerase (NEB #M0273) in a 40 µl final volume. PCR program was 95 °C 3 min; followed by 45 cycles of 95 °C 20 s, 62 °C 15 s and 72 °C 50 s, followed by 72 °C 5 min.

## Individual telomere sequencing by Nanopore
6R telomeres were amplified by TeloPCR with the following modifications. Primers used were oT2132 and oT2158. PCR mix was composed of 1× Colorless GoTaq Flexi buffer (Promega, Cat M7806) and 2.5 U of GoTaq G2 Flexi DNA polymerase. TeloPCR products were purified from a 1.5% agarose gel using the Macherey Nagel NucleoSpin Gel and PCR Clean-up Kit (Macherey Nagel #740609.50), and cloned into pUC18 amplified with oT2162 and oT2163 using the NEBuilder HiFi DNA Assembly following supplier instructions. 2 µl of the ligation was transformed into Stable competent bacteria (NEB #C3040) according to supplier instructions. Plates containing 500–10,000 colonies were scraped and plasmids were extracted in batch with the NucleoSpin Plasmid kit (Macherey Nagel #740588.50). -1 µg of plasmid DNA was linearized with Eco RI (NEB #R3101) and libraries of 12 samples were multiplexed using the Ligation sequencing gDNA−Native Barcoding Kit 24 V14 (ONT SQK-NBD114.24) and run on a MinION flowcell (v.10.4.1). For analyses, two distinct pipelines were followed. In Fig. 3d and Suplementary Fig. 3b–f, reads were basecalled and demultiplexed by Guppy v6.5.7. Nanopore adapter barcodes were removed and reads were demultiplexed by TelIndex. Resulting fastq reads displaying the same TelIndex were aligned using a custom Geneious Prime 2023.2.1 workflow (see Supplementary Pipeline 1). Resulting Consensus were aligned to eliminate residual minor sequencing errors manually and visualizations were made using Geneious Prime 2023.2.1. For Fig. 3e, f, Supplementary 3g, and Supplementary Pipeline 2 was followed.

## Microfluidics
Experiments were conducted as described[34] in 10-chamber chips (see Data Availability) casted with polydimethylsiloxane (PDMS; Sylgard 184) and curing agent mixed in a 10:1 ratio, degassed with a vacuum pump for 30 min, and poured into the mold. The PDMS was cured by baking for 5 h at 70 °C and then carefully removed from the mold. A biopsy puncher (1.5 mm; Harris Unicore) was used to create holes to connect the tubing. The PDMS and a glass coverslip (24 × 50 mm) were surface-activated by plasma (Diener Electronic) to covalently bond the two parts. YP 2% Raffinose or YP 2% Raffinose 3% Galactose dissolved in milliQ water were filtered using a 0.22-µm polyethersulfone filter (Millipore) and loaded into the device using a peristaltic pump (IPCN, Ismatec). Cells from a log-phase culture (OD$_{600nm}$ = 0.5) were injected into the device using a 1-mL syringe. A constant medium flow (10 µL/min) was maintained throughout the experiment and media replaced every 48 h. Cells were imaged using a fully motorized Axio Observer Z1 inverted microscope (Zeiss) with a 100× immersion objective, a Hamamatsu Orca R2 camera, and constant focus maintained with focus stabilization hardware (Definite focus, Zeiss). For phase contrast images, we used light-emitting diode light sources (Colibri 2, Zeiss) with the following

settings for standard microfluidic time-lapse experiments: 4.0 V for 70 ms. The temperature was maintained at 30 °C with a controlled heating unit and an incubation chamber that held the entire microscope base, including the stage and the objectives. Images were acquired every 10 min using Zen software (Zeiss). All aspects of image acquisition were fully automated and controlled, including temperature, focus, stage position, and time-lapse imaging. Time-lapse images were analyzed using a Matlab R2021b software as described[53]. Lineages that experienced very long cell cycles before the galactose addition were excluded from the analysis. The threshold for prolonged cell cycles was set to [mean + 2 × SD] of the dataset, the mean and SD being computed from cell cycle durations in media containing raffinose only or raffinose and galactose. For each set of experiments, at least two independents original transformants for the FinalCut constructs were involved in triplicate experiments.

## Mathematical modeling
The mathematical model is based on our previous work[30] and more details, original and FinalCut-adapted codes are available (see Code Availability section). The following modifications have been implemented to encode the FinalCut experimental setting. Time 0 in the simulation still corresponds to telomerase inactivation, but we now also model galactose addition at time 6 h. As soon as galactose is added, we forget the lineage's past history and reset cell generation so the first cell born under galactose has generation 0. From this time until time $t_{noCut} = t_{raf} + delay$, uncut cells have a certain probability at each division to have one of their telomeres cut down to a defined length $\ell_{cut}$. We chose $delay = 9$ h and $t_{raf} = t_{gal} + 6$ h. We then estimated the probability of cut from the cumulated distribution of cells where a telomere has been cut, fitted with the experimental cutting efficiency of FC20 (Fig. 1e).

Despite adding a probability to be cut, the implementation of galactose and raffinose conditions also comes with new distributions of cell cycle duration times. We followed the methodology of[30] to construct them, using the data of strain noFC noCas9 noDOX (Supplementary Fig. 4b). As indicated above, the threshold for prolonged cell cycles was set to [mean+ 2 × SD] of the dataset, the mean and SD being computed from cell cycle durations in media containing raffinose only or raffinose and galactose. Other parameters of the model required a few adjustments. We modified the parameter $\ell_{min,A}$ to $\ell_{min,A} = 36$ bp in order to better fit, for all $\ell_{cut}$ values, the generations of senescence onset obtained with FinalCut experiments. Because the parameters of $p_{sen.A}$, $p_{sen.B}$, and $p_{nta}$ are not independent, modifying $p_{sen.A}$ through $\ell_{min,A}$ cannot be done without slight corrections of the other parameters, in such a way that the previous experiments (simulated in ref. 30) are still well fitted (Supplementary Fig. 5b). These slight modifications are in accordance with our previous sensitivity simulations, which proved robustness of our fits to minor changes of the value of $\ell_{min,A}$.

## Fluctuation assay for mutation rate estimation
Independent transformants with pT126 plasmid of relevant strains (Supplementary Table 1) were inoculated in 50 mL of S -LEU R and 30 µg/mL doxycycline to obtain exponentially growing cultures after 16 h at 30 °C. 200 cells were plated on YPD plates (control) and $10^6, 10^7$, or $10^8$ cells were plated on synthetic complete medium-containing 2% glucose (SD) plates lacking arginine and supplemented with 60 µg/mL canavanine (Can plates). These cultures were considered as time zero. To induce Cas9-AID*-3MYC, time 0 cultures were diluted into 40 mL of S -LEU RG and 30 µg/mL doxycycline at an appropriate concentration to obtain an exponentially growing culture after 24 h. Cultures were counted and plated as described above. For Canavanine and 5-FOA double simultaneous selection experiments, independent clones were inoculated in 3 mL of S -LEU R medium and cultured overnight. The following day, cultures were diluted to an OD$_{600}$ = 0.05 in S -LEU R

medium supplemented with 30 μg/mL doxycycline and grown for 6 h. Each culture was then split into two: one received 15% (v/v) 20% galactose to induce expression; the other continued growth with raffinose as the sole carbon source. Both cultures were incubated to saturation for 2 days at 30 °C. Subsequently, 1 mL (~$10^8$ cells)−and, where applicable, multiples of 10 mL−were plated on Can plates also containing 5-fluoroorotic acid (5-FOA) at 1 g/L (w/v) as described[98]. For viability normalization, 100 μL of a $10^{-5}$ dilution of each culture was plated on YPD. Plates were incubated for 48 h at 30 °C, and colonies counted. The mutation rate was estimated as described[54] using a Matlab code (see Source Data File). For each set of experiments, at least two independent original transformants for the *FinalCut* constructs were involved unless otherwise stated (see Supplementary Tables 10 and 11 and Source Data Files).

## Pulse-field gel electrophoresis (PFGE)

For separation of yeast chromosomes, $5 \times 10^7$ cells from exponential cultures were washed in 10 mM Tris-HCl and 50 mM EDTA (pH 8.0) and then cast in 0.45% low melting point agarose (MP Biomedicals) plugs (molds, Biorad) in Zymoliase buffer (50 mM potassium phosphate, 50 mM EDTA, pH 7.0). Plugs were incubated in the Zymoliase buffer containing 10 mM DTT and 0.4 mg/ml Zymoliase 20 T (Seikagaku) overnight at 37 °C. The solution was replaced by the proteinase K buffer (10 mM Tris, pH 7.5, 50 mM EDTA, 1% Sarkosyl) and 2 mg/ml proteinase K (NEB), and then incubated 24 h at 50 °C. After washing three times with 10 mM Tris-HCl and 50 mM EDTA (pH 8.0), plugs were then inserted into the wells of a 0.9% agarose (Seakem Gold, Lonza) gel in 0.5× Tris-Borate-EDTA. Whole chromosomes were then separated in a rotating PFGE apparatus (Rotaphor 6.0, Biometra) with parameters chosen for *Saccharomyces cerevisiae* according to manufacturer instructions. DNA was visualized by staining with 0.5 μg/mL ethidium bromide under UV illumination (Gel Doc, Bio-Rad). High molecular weight genomic DNA size was estimated through an 0.75% agarose gel in 0.5× TBE in a PFGE apparatus (Rotaphor 6.0, Biometra) following parameters: pulse time 4–2 s (decreasing linearly), a rotor angle of 120–110° (decreasing linearly) and voltage of 130–90 V (decreasing linearly) during 18 h at 13 °C.

## Whole genome sequencing

High molecular weight genomic DNA was extracted from exponential phase cultures using MasterPure™ yeast purification kit (Biosearch technologies, #MPY80200) or Genomic DNA buffer set (Qiagen, #19060) with Qiagen Genomic-Tip 100/G (Qiagen, #10243), following the supplier instructions. DNA concentration was measured using Qubit 2.0 fluorometer with Qubit™ dsDNA BR Assay Kit (Invitrogen, #Q32853), while DNA purity was estimated by measuring the $OD_{260}/OD_{280}$ using NanoDrop™ 2000. DNA length was determined by PFGE. Libraries were prepared according to the Ligation sequencing gDNA−Native Barcoding Kit 24 V14 (ONT SQK-NBD114.24) and run on a MinION flowcell (v.10.4.1). The signal was basecalled using Dorado v0.71 and high accuracy model. Reads were mapped on the W303 (ura-can) genome and CGR are reported using LAST software (lastal v.1548) and dnarrange, respectively[99]. CGR were visualized using CIRCOS[100].

## Reporting summary

Further information on research design is available in the Nature Portfolio Reporting Summary linked to this article.

## Data availability

The microfluidics chip design and movies generated in this study have been deposited in Zenodo (https://doi.org/10.5281/zenodo.13821423)[101]. The sequencing data generated in this study have been deposited in the European Nucleotide Archive (ENA) at EMBL-EBI under accession number PRJEB80380. Source data are provided with this paper.

## Code availability

Codes for mathematical modeling are available at GitHub (https://github.com/anais-rat/telomeres)[102]. The Geneious Prime 2023.2.1 workflows for individual telomere sequence analysis, along with the MATLAB code for mutation rate calculation from fluctuation assays, are available on Zenodo (https://doi.org/10.5281/zenodo.13821423)[101].

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

## Acknowledgements

We wish to thank E. Fabre and V. Borde for sharing technical advice, Z. Xu for first mathematical simulations and fluctuation assay code and M. Godinho Ferreira, C. Azzalin, B. Llorente and G. Fischer for fruitful discussion. We also acknowledge J. Haber, H. Ulrich, M. Lisby for strains and plasmids and Virgile Andreani for improvements with codes. We also thank the Teixeira lab, numerous trainees and the UMR8226 unit members for technical support and fruitful discussions. We also thank T. Weinert, D. Kappei, E. Fabre, and M. Bao (Life Science Editors) for critical reading of the manuscript. P.B. was supported by a fellowship from the Ligue Contre le Cancer (France), F.R.R.B. was supported by a "Consejo Nacional de Ciencia y Tecnología" (CONACYT) scholarship and K.C. was supported by a fellowship from "Fondation pour la Recherche Médicale" (FRM). Work in M.C. lab was supported by an Open Competition M-2 grant from the Dutch Research Council. The work conducted in the MTT laboratory was funded by the Fondation pour la Recherche Médicale (FRM) through its "Équipe Labellisée" program and by the French National Research Agency (ANR) under the "Investissements d'Avenir" initiative (LabEx Dynamo, ANR-11-LABX-0011-01). The M.T.T. and M.D. laboratories also receive support from the ANR as part of the France 2030 program (23-EXMA-0005). Additionally, the M.T.T., G.C., and M.D. teams are funded by the ANR (ANR-16-CE12-0026) and the Institut National du Cancer (PLBIO20–312).

## Author contributions

Conceptualization Ideas; formulation or evolution of overarching research goals and aims: P.B. (Fig. 2), V.M.F. (Figs. 6 and 7), R.L. (Figs. 1 and 2), S.M. (Figs. 1 and 4), M.D. (Fig. 4), M.C. (Fig. 5), M.T.T. (Figs. 1–4, 6, and 7); Methodology Development or design of methodology; creation of models: P.B. (Figs. 1, 2, and 4), V.M.F. (Figs. 6 and 7), A.R. (Fig. 4), F.R.B. (Fig. 5), P.J. (Fig. 3), R.L. (Figs. 1–2, 4), S.M. (Figs. 1–2, 4), T.A. (Fig. 4), B.Z. (Fig. 4), K.C. (Fig. 6), H.G.K. (Fig. 5), GC (Fig. 4), M.D. (Fig. 4), M.C. (Fig. 6), M.T.T. (Figs. 1–3); Investigation Conducting a research and investigation process, specifically performing the experiments, or data/evidence collection: P.B. (Figs. 1–4), V.M.F. (Figs. 3, 6, and 7), A.R. (Fig. 4), F.R.B. (Fig. 5), P.J. (Fig. 3), B.Z. (Fig. 4), K.C. (Fig. 6), H.G.K. (Fig. 5), M.D. (Fig. 4); Data curation Management activities to annotate (produce metadata), scrub data and maintain research data (including software code, where it is necessary for interpreting the data itself) for initial use and later re-use: P.B. (Figs. 1–4), V.M.F. (Figs. 6 and 7), A.R. (Fig. 4), F.R.B. (Fig. 5), P.J. (Fig. 3), B.Z. (Fig. 4), M.D. (Fig. 4), M.T.T. (Fig. 3); Formal Analysis Application of statistical, mathematical, computational, or other formal techniques to analyse or synthesize study data: P.B. (Figs. 1–4), V.M.F. (Figs. 6 and 7), A.R. (Fig. 4), F.R.B. (Fig. 5), P.J. (Fig. 3), B.Z. (Fig. 4), M.D. (Fig. 4), M.T.T. (Fig. 3); Software Programming, software development; designing computer programs; implementation of the computer code and supporting algorithms; testing of existing code components: V.M.F. (Figs. 6 and 7), A.R. (Fig. 4), P.J. (Fig. 3), A.M. (Figs. 3 and 7), G.C. (Fig. 4), M.D. (Fig. 4), M.T.T. (Fig. 3); Validation Verification, whether as a part of the activity or separate, of the overall replication/reproducibility of results/experiments and other research outputs: P.B. (Figs. 1, 2, and 4), V.M.F. (Figs. 6–7), A.R. (Fig. 4), F.R.B. (Fig. 5), P.J. (Fig. 3), K.C. (Fig. 6), M.D. (Fig. 4), M.T.T. (Figs. 1–4, 6, and 7); Visualization Preparation, creation and/or presentation of the published work, specifically visualization/data presentation. P.B. (Figs. 1, 2, and 4), V.M.F. (Figs. 6 and 7), A.R. (Fig. 4), P.J. (Fig. 3), A.M. (Fig. 7), M.D. (Fig. 4), M.C. (Fig. 5), M.T.T. (Figs. 3, 5–7); Acquisition of the financial support for the project leading to this publication. P.B. (Figs. 1–4), K.C. (Fig. 6), G.C. (Fig. 4), M.D. (Fig. 4), M.C. (Fig. 5), M.T.T. (Figs. 1–4, 6, and 7); Resources Provision of study materials, reagents, materials, patients, laboratory samples, animals, instrumentation, computing resources, or other analysis tools. V.M.F. (Figs. 6 and 7), A.R. (Fig. 4), G.C. (Fig. 4), M.D. (Fig. 4), M.C. (Fig. 5), M.T.T. (Figs. 1–4, 6, and 7); Project administration Management and coordination responsibility for the research activity planning and execution. V.M.F. (Figs. 6 and 7), R.L. (Figs. 1, 2, and 4), S.M. (Figs. 1 and 4), G.C. (Fig. 4), M.D. (Fig. 4), M.C. (Fig. 5), M.T.T. (Figs. 1–7); Supervision Oversight and leadership responsibility for the research activity planning and execution, including mentorship external to the core team. V.M.F. (Figs. 6 and 7), R.L. (Figs. 1, 2, and 4), S.M. (Figs. 1 and 4), G.C. (Fig. 4), M.D. (Fig. 4), M.C. (Fig. 5), M.T.T. (Figs. 1–4, 6, and 7); Writing–original draft Preparation, creation and/or presentation of the published work, specifically writing the initial draft: P.B., V.M.F., S.M., M.T.T.; Writing–review & editing Preparation, creation and/or presentation of the published work by those from the original research group, specifically critical review, commentary or revision–including pre- or post-publication stages. P.B., V.M.F., A.R., F.R.B., P.J., R.L., S.M., A.M., T.A., B.Z., K.C., H.G.K., G.C., M.D., M.C., M.T.T.

## Competing interests

The authors declare no competing interests.
