## [Transparent Peer Review file · Nature Communications]

Both Genome Instability and Replicative Senescence Stem from the Shortest Telomere in Telomerase-Negative Cells

Corresponding Author: Dr Maria Teresa Teixeira

Version 0:

Reviewer comments:

Reviewer #1

(Remarks to the Author)

This study addresses the longstanding question of what exactly is the minimal trigger for telomere-driven senescence. The authors investigate this by taking advantage of detailed single-cell and molecular analysis in the budding yeast system. To this end they developed a novel system, based on CRISPR-induced double-stranded breaks, to generate novel telomeres of defined lengths. The system is used in a series of careful experiments that link the length of the newly-exposed telomeric sequences to their ability to stimulate telomerase action or elicit DNA damage checkpoint activation. The length of newly-synthesised ends and the extent of shortening of the exposed ends is analysed in fine detail using deep sequencing. Importantly, a correlation is found between the predicted timing of telomere degradation at the novel telomere and the onset of senescence, as monitored by single-cell observation of cycling cells. This leads the authors to suggest that a single shortened telomere of length of about 30-40 bp is sufficient to trigger senescence in yeast cells. As this is simple correlative evidence, the authors model telomere shortening mathematically and estimate that the threshold length to trigger senescence is at 33-39 bp. This analysis is complicated by the fact that cells senesce according to two modes, only of which appears to follow a two-step partly stochastic mode. Although some variability in the onset of senescence is retained in this experimental set up, I think that the evidence linking this cellular outcome to the length of the single shortened telomere is compelling, and constitutes a significant advance. The work is beautifully designed and executed and is an important contribution to the general area of telomere dysfunction and senescence activation.

Minor points:

I think that the manuscript could be substantially shortened and figures combined; in my view the paper would benefit from simplifying its rather straightforward message. The last part on genome instability after telomere shortening is interesting but not crucial, and in my view unsurprising: I am not suggesting it should be eliminated, but I think its presentation could be streamlined and figures simplified.

I would consider combining Figure 4 and 5, and panels 7a/d and 7b/e, for example.

The authors address the question of whether the identified length threshold applies to endogenous telomeres besides the novel one at VI-R. They do so by identifying telomerase elongation events in cells where short telomeres have artificially shortened by limiting telomerase RNA expression. One possible caveat of these experiments is that the abnormal cycling conditions of these cells might have led to adaptive changes that make the cells more tolerant of shorter lengths. In any case, I think it is already well documented - including by the original telomere PCR experiments by the lead author - that telomerase elongation events are detected on very short telomeres.

I found the analysis of telomere shortening upon CRISPR cleavage less than ideal (Figure 3 and related data). The experimental design chosen ('enhanced') was designed to achieve a limited burst of CRISPR activity, followed by analysis of telomere length at telomere VI-R (degradation of Cas9 is even nicely included). A time 0 is set, but this is a bit misleading, since clearly not all the DSBs at VI-R will be t=0 ones, as some will have undergone early cleavage and then 1 or possibly 2 cell cycles. It would have been far preferable, in my view, to keep cells arrested with alpha factor until time 0. This would have made the analysis much more powerful, as it would have allowed to link observed telomere shortening to predicted cell division events. Having this type of knowledge about the number of cell division would presumably generate tighter classes

of telomere lengths and allow better inferences about the end replication problem. Even then I would find it problematic to use some of the language used in the paper. For example, at line 268 it is written that the 'data SUGGEST that FC50 shortens at a rate of ~6-11 bp on HALF of telomeres per cell division': this conclusion is based on previous models of how the end replication problem might play out, but there is nothing here to offer specific support for those models. I would recommend much more nuanced language ('suggest' is too strong for me here).

Just a few lines below (line 281), it is then stated that sequencing of the novel VI-R telomeres 'SHOWED that a single telomere of fixed length shortens in an asymmetrical manner': I did not find data that showed this in this manuscript, nor a convincing discussion to this regard.

The depiction of the authors' previous model for senescence, which envisions Type A and B cells, should be included in the main manuscript, and not just confined to the Supplements.

Although the Strecker et al paper is cited, I think it should be discussed more prominently given that it reached similar conclusions with regard to the minimal telomeric sequence that distinguishes a telomere from a DSB.

In Figure 1c, there are not size markers for the gel. Why are the cut fragments apparently the same size even though the TG stretch varies from 20 to 50 bp? Is there a corresponding random sequence filler, or is it lack of resolution?

In Figure 1d, the FC30 cut fragment increases in size from 3 to 9 hrs: this seems to indicate residual telomerase activity.

Lines 158-160. I find this statement inaccurate and not matching the quantification reported in the table. Perhaps the authors mean to say that about 90% of the uncut smear disappears at 9 hrs? Not at 6 hrs, and not all of it is certainly 'converted' to a sharp cut band as stated.

Line 323: it is stated here that the novel telomere is 'always the shortest one': is that demonstrated? It would seem conceivable that there could be cells with rare really short telomeres. For example the noFC control in Supplementary Figure 4 shows a few cells that senesce really quickly, presumably because carrying very short telomeres.

Line 228: I was surprised that only 'nearly half' the telomeres sequenced were found to be elongated, as Figure 1c seemed to indicate much more prominent elongation. Perhaps more context could be given?

Perhaps I missed it, and it is explained in detail somewhere, but what exactly is 'Telindex'?

Could the data in figure 4a be summarised quantitatively in the figure? Mean values of the distributions are given in the text but not in the figure.

The model in figure 5a is not sufficiently described. What do the different coloured lines represent? It also stated that cells with the the longer telomere 'seeds' displayed larger variation in cell cycle length: could this be quantified?

(Remarks on code availability)

Reviewer #2

(Remarks to the Author)

The authors present a new genetic approach to study the effects of altered telomere length in budding yeast, claiming it provides "precise experimental control of telomere length". It is an elegant approach that seems effective and, as the authors claim, allows them to examine the effects of a conditional shortening of a single telomere to a defined length. The central claim of the paper appears both compelling and directly tested. Overall, I found the paper to be clear, logical, and the experimental work carefully conducted/controlled. However, I am not an expert in telomere biology (more yeast genetics, cell cycle, genome biology), and so I am less well placed to comment whether their set-up is really the first time these questions can be tested directly, as they claim.

Comments for the authors:

1. The authors demonstrate that a threshold telomere length can induce senescence, but I am less convinced that this does happen in any physiological context. My understanding is that, unlike mammalian cells, telomere shortening is not generally understood to be the primary driver of senescence in wild-type yeast (for which their RNRpr-TLC1 strain is unlikely to be representative). I think it would significantly help the author's claim if they could perform additional experiments to detect naturally short telomeres in a population enriched for naturally aged mother cells.

2. While 2-3 replicates blots of most experiments are quantified, it appears that the claim that the degradation of FC20 and FC30 extends internally is just based on one blot (Supp. Fig. 1D). Replicates and quantifications should be added. It also appears that where replicate blots have been performed and quantified, only one blot is shown - ideally, all blots should be shown in the supplement.

3. The multiple references to oncogenesis in the abstract seem a stretch given the work's exclusive use of a single-celled

microbe to study the effects of telomere shortening. I do not think that the use of the model systems in any way counts against the value of the work (in fact, its value lies in taking advantage of the systems' strengths). However, I feel the authors need to tone down their language to reflect the conclusions that their data directly support.

(Remarks on code availability)

Reviewer #3

(Remarks to the Author)

Review Manuscript NCOMMS-25-35131-T

Summary:

In the manuscript entitled "Both Genome Instability and Replicative Senescence Stem from the Shortest Telomere in Telomerase-Negative Cells" Berardi P. et al combine yeast genetics with single-cell multifluiddic lineage tracing, modeling and whole-genome and telomere sequencing to address the consequences of one short telomere on genome stability and proliferation in *Saccharomyces cerevisiae*. The engineered FinalCut system allows the authors to investigate the cellular consequences of one short telomere of defined lengths at chromosome 6R in high resolution. This system is superior to previous telomere dysfunction models as it allows for precise temporal and telomere length control at a native subtelomeric context and thereby overcomes the intrinsic intracellular telomere heterogeneity. The analysis suggests that in budding yeast telomeres shorter than 30-40 bp become dysfunctional and cause replicative senescence and genome instability. Their work elegantly shows that one short dysfunctional telomere is sufficient to cause replicative senescence in budding yeast. Further, their data indicate that genome instability induced by one dysfunctional telomere acts predominantly in cis. To comprehensively address the impact of one critically short telomere on genome instability in trans further investigations will be needed.

Together, the work provides important mechanistic insights on how one single short telomere restricts eukaryotic proliferation capacity and causes genome instability. If the proposed model is conserved between other eukaryotes including human, the work has important clinical consequences for short telomere syndrome patients and Alternative Lengthening of Telomeres. In the future, it will be interesting to apply the developed methodology to investigate which other changes next to the telomere sequence loss occur at short telomeres and drive telomere dysfunction as well as the mechanisms that contribute to the observed genome instability.

Major comments:

- 1) The authors suggest that degradation of FC20 and FC30 may extend internally into the chromosome using a more internal TEL6R fragment. However, the representative blot shown in Supplementary Fig. 1d shows also reduced signal in the later time points of the FC70 sample. Further, what is the proposed mechanism for the loss of signal: nuclease-dependent degradation or terminal loss due to a GCR event? Can the authors please clarify this point?
- 2) In Fig. 2 b-c, the authors investigate the kinetics of telomeric signal loss at FinalCut Telomeres of varying length. Based on the representative blots the loss of signal seems to be dependent on the length of the FCx fragment with faster kinetics in FC40 relative to FC50. However, in the quantification, FC40 and FC50 have almost similar kinetics, especially in the first 2 h.
 - a. Can the authors please comment on this?
 - b. Can the authors please specify why in some samples 4 or 5 individual data points are shown, even though the legend says that the average of three experiments is shown?
 - c. As telomere shortening due to the end replication problem is in large part replication-dependent, can the authors please provide the corresponding population doubling to the indicated time points?
- 3) The sequencing analysis of FC50 and FC70 revealed an average telomere shortening of 4.4 and 3.4 bp/PD, respectively (Fig. 3e). Can the authors please perform linear regression analysis and provide statistics using all three time points? Is there a biological reason why longer telomeres seem to shorten slower?
- 4) Interestingly, the authors observed that 23.8% +/- 2.2 FinalCut telomeres shorten by 6 bp and end in position 51 (Fig. 3f), suggesting that those telomeres undergo telomere shortening prior to sampling. Is the shortening/observed telomere length at 0 hours dependent on when in the cell cycle the FinalCut was performed?
- 5) With increasing FC, the median PD before senescence onset increases (Fig. 4b and Supplementary Fig. 4). The author could calculate the telomere shortening rate per PD by plotting FC length against the median PD before senescence onset and performing linear regression analysis. Assuming linear shortening how long would the estimated shortest telomere in yeast with heterogeneous telomere length that enter senescence after median 30 PDs be? How does the length of the predicted shortest telomere relate to individual telomere length measurements in wild-type yeast?
- 6) Can the authors please change their color code for some of their figures to allow better readability for people with color blindness (e.g. change red-green to red-blue in Fig. 4 and Supplementary Fig. 4)?
- 7) To test whether short telomeres of 30-40 bp identified with FinalCut can also be detected at native telomeres the authors make use of a PRNR3-TLC1 strain and sequence individual 6R telomeres. Chromosome arm-specific telomere length sequencing of Sholes SL et al (PMID: 34702734) suggests that the telomere length at chromosome 6R is quite average.

This would predict that the shorter telomeres at other chromosome arms trigger the checkpoint response and consequently telomerase activity earlier. Why have the authors sequenced 6R telomeres and not for example the overall shorter 1L or 11L telomeres, so that the checkpoint activity can be more directly connected to the short telomere? Which outcome do the authors expect in case of sequencing the chromosome arms with the overall shorter telomeres?

8) The authors conclude based on their CAN1/GCR fluctuation assays that the genome instability during replicative senescence occurs primarily in cis (Fig. 7, Supplementary Fig 7 and Supplementary Table 9). This is in line with the presented data and it is intuitive that the arm with the shortest telomere is the one that is most genetically unstable. However, I have several comments and questions:

- a. Why are the authors using sequential drug treatment and not simultaneous canavanine and 5-FOA selection to investigate GCR? I have two remarks here: First, the additional growth time may result in more double-negative cells. Second, CAN1 (and URA3) can be inactivated by several mechanisms (base substitutions, insertions/deletions, GCRs,...). In wild-type budding yeast CAN1 is mainly inactivated by base substitutions and smaller INDELs (PMID: 9702187, 12972632, 20961955), and infrequently by GCRs. So, if the CAN1 mutation rate ($\sim 10^{-7}$, mainly driven by base substitutions and INDELs) is orders of magnitude higher than the GCR rate ($\sim 10^{-10}$ for classical GCR assay, PMID: 11239397), it is not surprising that few GCR are detected in trans in the canavanine resistant colonies. The GCR assay in cis detects GCRs because the GCR rate is increased due to the adjacent critically short telomere. Consequently, the increased GCR rate in cis is comparable or higher than the CAN1 mutation rate and GCR are detected with sequential drug treatment. Importantly, it may be that the critically short telomere at 6R causes an increase in GCR rate in trans, even though at lower levels than in cis, but this increase is “masked” by the more frequent CAN1 inactivation events and the assay setup. Can the authors please measure GCR rates with simultaneous canavanine and 5-FOA selection?
- b. Based on the presented sequencing of GCR+ colonies resulting in loss of the CAN1-URA3 cassette upon FinalCut in can1 cells, the authors conclude that non-reciprocal translocations are the signature of shortest telomeres. The authors could modify their GCR strain like Putnam et al. (PMID: 19641493) and integrate an antibiotic cassette telomeric to the introduced CAN1-URA3 cassette on 6R. This way they could simultaneously measure GCRs on canavanine-5-FOA containing media and quantify how many GCR events are indeed due to the chromosome arm loss.
- c. As the authors observe stimulated by the homology of HXK1 and HXK2 non-reciprocal translocation as a prime GCR event. It will be interesting to delete either HXK1 or HXK2 and perform GCR assays (with simultaneous canavanine and 5-FOA treatment) in cis and trans. The absence of the homology site may result in more complex GCRs and increased in trans genome instability.
- d. The authors speculate that Exo1, Rad1 and BIR are involved in the formation of FinalCut-induced GCRs. To clarify the mechanism, can the authors please measure GCR rates in their FinalCut can1 GCR strain in the absence of Exo1 and Rad1, as well as in strains lacking Rad52, Rad51, Rad59 and Pol32 combination thereof?
- e. The authors conclude that the shortest telomere is causing genome instability primarily in cis. This interpretation is based mostly on measured CAN1/GCR rates with the CAN1-URA3 cassette integrated either at chromosome arm 5L or 6R. While being a powerful assay the GCR assay only scores for quite specific events – loss of the CAN1-URA3 cassette and sufficiently protected chromosome ends and growth so that the event can be counted as a colony. For a critically short 6R telomere to cause an increased GCR rate in trans at 5L, the 6R arm would likely need to fuse with 5L, the fused chromosomes break resulting in loss of the CAN1-URA3 cassette and the broken ends stabilized by telomeres. This is a much more complex chain of events than a GCR in cis. Further, the critically short 6R telomere could in principle fuse with 5L and 30 other chromosome ends. This alone would suggest that the GCR rate at 5L is at least approximately 30x lower than the GCR rate than at 6R (assuming that all chromosome arms are equally fusion-prone). Thus, to investigate the possible genome instability impact of one critically short telomere at 6R in trans the authors could do two experiments:
 - i. Measure GCR rates (see above 8a-c).
 - ii. Measure 6R chromosome fusions with other chromosome arms in a fusion PCR assay like assays performed in mammalian cells by the Baird lab (PMID: 17908935).

9) The possibility that one single short telomere is necessary and sufficient also in other eukaryotes is an exciting hypothesis. One major difference between the used budding yeast strains and most other eukaryotic systems is, that the used yeast strains are haploid, whereas most eukaryotic systems are diploid. It will be interesting to investigate heterozygous or homozygous FinalCut diploid yeast and the impact of critical short telomere(s) in diploid yeast.

Minor comments:

- 1) Page 8 line 220-221: please add SacI site and I am wondering whether the “a common stretch of degenerated TG1-3 repeats”, is redundant to “followed by TG1-3 of variable length and divergent sequence among the telomere molecules”?
- 2) Page 8 line 228-229: In Supplementary Fig. 3c can you please indicate which of the telomeres experienced de novo telomere re-elongation?
- 3) Page 12 line 331: How is the “long” cell cycle threshold defined? Dependent on the media the ratio is 2x and 1.6 fold? Please explain.
- 4) In Supplementary Fig. 3 b-f the authors could mark/highlight the regions like FC50, and the ttDNA, similar to Fig. 1a.
- 5) Can you please add as comparison Supplementary Fig. 4d to Fig. 4b? Can the author please also indicate in Fig. 4b and the Supplementary Fig. 4 panels the median cell divisions before senescence onset?
- 6) In some figures the light blue/turquoise text is very light and difficult to read if printed (e.g Fig. 1, Fig. 4b FC20; Supplementary Fig. 1 FC0 and FC20; 5b FC0). Can you please change it to a color that is more visible?

7) For the mutation rates please check number of cultures (Supplementary Table 9). Please add some additional cultures as described in methods for the ones that have very few cultures.

8) Supplementary Fig. 6 is missing. Please adjust the numbering of Supplementary Figures accordingly.

9) Please use either "." or "," throughout the manuscript e.g. Supplementary Table 5 pH7.2 or pH7,4.

(Remarks on code availability)

Version 1:

Reviewer comments:

Reviewer #1

(Remarks to the Author)

I am satisfied that the revised manuscript addressed my comments (although I would like the authors to still clarify in the new figure 4b what the different shades of green exactly represent).

(Remarks on code availability)

Reviewer #2

(Remarks to the Author)

The authors have improved the manuscript through the revision process, and I am happy they have adequately dealt with the questions/points that I raised.

(Remarks on code availability)

Reviewer #3

(Remarks to the Author)

My main comments have been addressed in the revised manuscript of this interesting work. I especially appreciate the author's work on GCR fluctuations and pol32/exo1/rad1 knock-out strains.

(Remarks on code availability)

Reviewer #1 (Remarks to the Author):

This study addresses the longstanding question of what exactly is the minimal trigger for telomere-driven senescence. The authors investigate this by taking advantage of detailed single-cell and molecular analysis in the budding yeast system. To this end they developed a novel system, based on CRISPR-induced double-stranded breaks, to generate novel telomeres of defined lengths. The system is used in a series of careful experiments that link the length of the newly-exposed telomeric sequences to their ability to stimulate telomerase action or elicit DNA damage checkpoint activation. The length of newly-synthesised ends and the extent of shortening of the exposed ends is analysed in fine detail using deep sequencing. Importantly, a correlation is found between the predicted timing of telomere degradation at the novel telomere and the onset of senescence, as monitored by single-cell observation of cycling cells. This leads the authors to suggest that a single shortened telomere of length of about 30-40 bp is sufficient to trigger senescence in yeast cells. As this is simple correlative evidence, the authors model telomere shortening mathematically and estimate that the threshold length to trigger senescence is at 33-39 bp. This analysis is complicated by the fact that cells senesce according to two modes, only of which appears to follow a two-step partly stochastic mode. Although some variability in the onset of senescence is retained in this experimental set up, I think that the evidence linking this cellular outcome to the length of the single shortened telomere is compelling, and constitutes a significant advance. The work is beautifully designed and executed and is an important contribution to the general area of telomere dysfunction and senescence activation.

We thank this reviewer for this positive comment.

Minor points:

I think that the manuscript could be substantially shortened and figures combined; in my view the paper would benefit from simplifying its rather straightforward message. The last part on genome instability after telomere shortening is interesting but not crucial, and in my view unsurprising: I am not suggesting it should be eliminated, but I think its presentation could be streamlined and figures simplified.

I would consider combining Figure 4 and 5, and panels 7a/d and 7b/e, for example.

Figures 4 and 5 were combined and Figure 7 was modified according to Reviewer #1 and #3 suggestions.

In addition, in the updated version of our manuscript, we have substantially expanded the section on genome instability to better highlight the novelty of our findings. While it is established that genomic instability increases in telomerase-negative populations as senescent cells accumulate, our work now clearly demonstrates two key points:

- 1- **Origin of Instability:** We show that genomic instability arises specifically from cells harboring a single critically short telomere at the point of dysfunction.
- 2- **Localized Effect:** This instability predominantly accumulates in cis with the short telomere — a scenario consistent with, but not previously directly demonstrated in, existing literature.

This direct evidence not only clarifies the mechanism but also distinguishes our findings from prior observations.

The authors address the question of whether the identified length threshold applies to endogenous telomeres besides the novel one at VI-R. They do so by identifying telomerase elongation events in cells where short telomeres have artificially shortened by limiting telomerase RNA expression. One possible caveat of these experiments is that the abnormal cycling conditions of these cells might have led to adaptive changes that make the cells more tolerant of shorter lengths. In any case, I think it is already well documented - including by the original telomere PCR experiments by the lead author - that telomerase elongation events are detected on very short telomeres.

We agree that it is formally possible that abnormal cycling conditions might have led to adaptive changes that make cells more tolerant to shorter telomere lengths. Actually, the phenomena of adaptation to a DSB is part of the process of replicative senescence as described in Coutelier et al. *Genes & Dev* 2018 (DOI: 10.1101/gad.318485.118). So, both experimental situations would be similar. Nevertheless, as the reviewer notes, prior telomere PCR experiments also support the notion that telomerase is able to elongate very short telomeres.

I found the analysis of telomere shortening upon CRISPR cleavage less than ideal (Figure 3 and related data). The experimental design chosen ('enhanced') was designed to achieve a limited burst of CRISPR activity, followed by analysis of telomere length at telomere VI-R (degradation of Cas9 is even nicely included). A time 0 is set, but this is a bit misleading, since clearly not all the DSBs at VI-R will be t=0 ones, as some will have undergone early cleavage and then 1 or possibly 2 cell cycles. It would have been far preferable, in my view, to keep cells arrested with alpha factor until time 0. This would have made the analysis much more powerful, as it would have allowed to link observed telomere shortening to predicted cell division events. Having this type of knowledge about the number of cell division would presumably generate tighter classes of telomere lengths and allow better inferences about the end replication problem.

The experiments described in Fig. 3 were designed to verify that *FinalCut* behaves as a native telomere within the established framework of the DNA end replication problem. We agree with the reviewer that the current experimental system is underexploited, as it presents exciting opportunities for deeper investigation into the DNA end replication problem itself.

To pursue these opportunities, further methodological developments will be required, including reconstructing relevant strains in a *MATa* background to enable alpha-factor synchronization and adapting the experimental conditions accordingly. Key challenges will include optimizing Cas9 activity during prolonged G1 arrest and ensuring robust progression into S phase following release. These important extensions will be addressed in a dedicated future study in our laboratory.

Even then I would find it problematic to use some of the language used in the paper. For example, at line 268 it is written that the 'data SUGGEST that FC50 shortens at a rate of ~6-11 bp on HALF of telomeres per cell division': this conclusion is based on previous models of how the end replication problem might play out, but there is nothing here to offer specific support for those models. I would recommend much more nuanced language ('suggest' is too strong for me here).

Just a few lines below (line 281), it is then stated that sequencing of the novel VI-R telomeres 'SHOWED that a single telomere of fixed length shortens in an asymmetrical manner': I did not find data that showed this in this manuscript, nor a convincing discussion to this regard.

We agree with the reviewer and acknowledge the fact that much of the DNA end replication problem remains to be investigated. Text was modified to nuance the statements:

- Line 268 we changed "suggest" by "is compatible with a model where"

- Line 281: we removed “in an asymmetrical manner”, which is not required for our main conclusion there and subsequent sections of the manuscript.

The depiction of the authors’ previous model for senescence, which envisions Type A and B cells, should be included in the main manuscript, and not just confined to the Supplements.

We agree with this reviewer that understanding the differences between type A and type B cells is important to understand the mathematical modeling. We have thus included the figure in the main manuscript (new Fig. 4c).

Although the Strecker et al paper is cited, I think it should be discussed more prominently given that it reached similar conclusions with regard to the minimal telomeric sequence that distinguishes a telomere from a DSB.

It is critical to highlight that the findings of Strecker et al. were derived from telomerase-positive cells, where the identified telomere length threshold is directly tied to telomerase activity. Telomerase’s capacity to distinguish natural telomeres from accidental double-strand breaks (DSBs) underpins this threshold.

In contrast, our work investigates the establishment of senescence in the absence of telomerase. Under these conditions, the telomere length threshold at which telomeres are recognized as accidental DSBs differs significantly. While telomerase can re-elongate telomeres above 30 bp, telomeres shorter than 40 bp in telomerase-negative cells become susceptible to degradation.

Notably, in telomerase-positive cells, telomeres rarely reach such critically short lengths. Telomerase is preferentially recruited to the shortest telomeres in wild-type cells, maintaining them well above 30–40 bp. However, when a 30 bp telomere is experimentally introduced, telomerase acts on it. This rapid response suggests that telomerase activity may outcompete DNA repair pathways, as suggested by our recent work (Rosas Bringas et al., 2024 DOI : 10.1073/pnas.2407314121) thereby preventing the extensive degradation that would otherwise occur in its absence.

To clarify this point we added in main text the following:

“Therefore, while telomerase can re-elongate telomeres above 30 bp, as previously described⁴², we find that telomeres shorter than 40 bp in telomerase-negative cells become susceptible to degradation. This suggests that telomere structures enabling telomerase recruitment and telomere protection might differ substantially.”

In Figure 1c, there are not size markers for the gel. Why are the cut fragments apparently the same size even though the TG stretch varies from 20 to 50 bp? Is there a corresponding random sequence filler, or is it lack of resolution?

We have now included size markers for the blot presented in Supplementary Fig 1b, c. We can appreciate that there would be very little resolution for 10 bp differences. There is no sequence filling to equalize electrophoretic mobility.

In Figure 1d, the FC30 cut fragment increases in size from 3 to 9 hrs: this seems to indicate residual telomerase activity.

Indeed, this a possibility, given the fact that most cells die after the generation of FC30, a strong selection for the few cells expressing small amounts of telomerase is expected to apply. Note that this re-lengthening is absent from FC20, in which telomerase is unable to re-elongate, as demonstrated in Strecker et al., mentioned above. We added a sentence:

« A minor subset of *FinalCut* telomeres above 30 bp exhibited lengthening. This phenomenon likely arises from sporadic *P_{telO2}* leakage in a subset of cells and selection, given that no such elongations were observed in the *FC20* strain, as expected for telomerase not recognizing a 20 bp-telomeric sequence as a substrate⁴². »

and changed the following sentence to

« Overall, our results indicate that most of the shorter *FinalCut* telomeres of *FC20* and *FC30* strains are likely degraded immediately, whereas the ones of *FC40* and *FC50* are initially stable prior to delayed degradation (Fig. 1f). »

Lines 158-160. I find this statement inaccurate and not matching the quantification reported in the table. Perhaps the authors mean to say that about 90% of the uncut smear disappears at 9 hrs? Not at 6 hrs, and not all of it is certainly 'converted' to a sharp cut band as stated.

Indeed there are some variations among the experiments and the constructs (ex: FC40 and FC50). We have corrected the statement to

"When telomerase was inactivated prior to Cas9 induction by doxycycline addition to the media to suppress *TLC1* expression, approximately 80-90% of the uncut smear signal was converted to a sharp cut band within 6-9 hours."

Line 323: it is stated here that the novel telomere is 'always the shortest one': is that demonstrated? It would seem conceivable that there could be cells with rare really short telomeres. For example the noFC control in Supplementary Figure 4 shows a few cells that senesce really quickly, presumably because carrying very short telomeres.

This is a highly pertinent question related to the longstanding issue of telomere length distribution in cells, which we specifically started to address in the Xu et al. Genetics 2013 (DOI: 10.1534/genetics.113.152322). Later on, in the Rat et al. Nat Comm 2025 (DOI: 10.1038/s41467-025-56196-z, look into Fig. 5a), we show that the shortest telomere in these cells (in which telomerase telomerase RNA is under the control of a conditional promoter) can reach up to 100 bp, and this is compatible with quick senescence in a few cells (see Fig. 2c of Rat et al. Nat Comm 2025 (DOI: 10.1038/s41467-025-56196-z for an in silico microfluidics experiment using this exact distribution of telomere length).

In the current experimental setting, we deplete telomerase activity prior to FinalCut induction. We have now proceeded to simulations using our mathematical model to estimate the fraction of cells in which the shortest telomere is actually the FinalCut we induce with our system and we obtain over 1000 simulations:

Supplementary Table 7: Fraction of cells in which the shortest telomere is the FinalCut telomere

FinalCut	Fraction of lineages in which the shortest telomere after the cut is FinalCut
FC20	1 (78531 / 81000)
FC30	1 (66047 / 68000)
FC40	1 (61191 / 63000)
FC50	1 (54244 / 56000)
FC70	0.999896773127980 (48437 / 50000)

Typically, at the time of FinalCut induction, the distribution of telomere length is the following (New Supplementary Fig. 5a):

Telomere length distribution at the time FinalCut is induced, as simulated for 1000 lineages with the mathematical model set for indicated FinalCut conditions.

This strongly suggests that FC is the shortest one in a large majority of cells. We now included this information and added in the text:

“In these conditions, our simulations confirm that in nearly all cells, the FinalCut telomere is actually the shortest telomere (Supplementary Fig 5a, Supplementary Table 7).”

Line 228: I was surprised that only ‘nearly half’ the telomeres sequenced were found to be elongated, as Figure 1c seemed to indicate much more prominent elongation. Perhaps more context could be given?

We appreciate the reviewer’s careful attention to this observation. Actually, the clarity of the FC50 lanes varied across independent experiments. Below, we provide an additional representative experiment in which the FC50 lanes are more clearly resolved. While this dataset was not included in the original figure due to the absence of the FC70 condition, we believe it effectively addresses the reviewer’s concern.

In this figure, we observe that the FC30 telomere appears fully elongated by telomerase, whereas the FC40 and FC50 telomeres exhibit only partial elongation, suggesting a differential regulation. While our

current experimental framework does not allow us to definitively explain this pattern, one plausible hypothesis is that, for FC40 and above, telomerase may preferentially elongate leading-strand telomeres, as previously proposed by Faure et al. (Mol Cell 2010; DOI: 10.1016/j.molcel.2010.05.016). Addressing this hypothesis directly will be an important focus of our future investigations.

Perhaps I missed it, and it is explained in detail somewhere, but what exactly is 'Telindex'?

We regret not having sufficiently explained the importance of TelIndex. TelIndex is a 10-nucleotide random sequence incorporated into the forward primer (oT2132, see Supplementary Table 4) during TeloPCR to uniquely barcode individual telomere molecules in genomic DNA extracted from cells. As a result, each telomere molecule—and its corresponding amplicons, especially during bacterial growth—is labeled with a unique identifier defined by this 10-mer sequence. The goal is to trace multiple Nanopore reads back to their original telomere molecule, allowing us to obtain read depth for each individual molecule. By grouping reads with identical TelIndex sequences, we can generate a consensus sequence, thereby improving accuracy down to nucleotide resolution. To aid reader understanding, we have modified Supplementary Fig. 3a to clarify this point.

Could the data in figure 4a be summarised quantitatively in the figure? Mean values of the distributions are given in the text but not in the figure.

Fig4a in the submitted paper corresponds to microfluidics experimental setting and contains no data. We regret not understanding to which figure this reviewer refers to.

The model in figure 5a is not sufficiently described. What do the different coloured lines represent? It also stated that cells with the the longer telomere 'seeds' displayed larger variation in cell cycle length: could this be quantified?

We regret not having been clearer in explaining the rationale behind our mathematical modeling. We have revised the figure 5a, now new figure 4b, next to the new Fig 4e, to address the reviewers' comments, and we hope this has now clarified the point. See below.

b, Description of the two routes to senescence found in microfluidics experiments of control telomerase-negative cells³⁴. Type A cells undergo senescence in a single abrupt transition from normal cell cycles (green) to terminally prolonged cell cycles (magenta). Type B cells undergo series of prolonged cell cycles, followed by normal cell cycles (dark magenta), before ending their route by terminally abnormally long cell cycles (magenta).

e, Mathematical model of telomerase-negative individual cells recapitulating Type A and Type B trajectories. Inset: Simplified tree diagram of Type A and Type B routes shown in b. Colors reflect cell cycle durations as in b. The graph shows the probability of senescence of Type A cells as function of the length of the shortest telomere ($p_{sen,A}(L)$) (dark red); the probability of Type A cells to become Type B - when undergoing a first non-terminal prolonged cell cycle -

as function of the length of the shortest telomere ($p_{nta}(\ell)$) (dark magenta); the probability of senescence of Type B cells as function of the length of the shortest telomere ($p_{sen,B}(\ell)$) (orange). Parameters listed in Supplementary Table 8.

Reviewer #2 (Remarks to the Author):

The authors present a new genetic approach to study the effects of altered telomere length in budding yeast, claiming it provides “precise experimental control of telomere length”. It is an elegant approach that seems effective and, as the authors claim, allows them to examine the effects of a conditional shortening of a single telomere to a defined length. The central claim of the paper appears both compelling and directly tested. Overall, I found the paper to be clear, logical, and the experimental work carefully conducted/controlled. However, I am not an expert in telomere biology (more yeast genetics, cell cycle, genome biology), and so I am less well placed to comment whether their set-up is really the first time these questions can be tested directly, as they claim.

We thank this reviewer for this positive comment.

Comments for the authors:

1. The authors demonstrate that a threshold telomere length can induce senescence, but I am less convinced that this does happen in any physiological context. My understanding is that, unlike mammalian cells, telomere shortening is not generally understood to be the primary driver of senescence in wild-type yeast (for which their RNRpr-TLC1 strain is unlikely to be representative). I think it would significantly help the author’s claim if they could perform additional experiments to detect naturally short telomeres in a population enriched for naturally aged mother cells.

This manuscript focuses solely on telomere shortening–dependent replicative senescence, not on mother cell aging—a separate form of cellular senescence in budding yeast. Current evidence suggests these processes are driven by distinct mechanisms, as telomeres do not shorten during mother cell aging (D’Mello & Jazwinski, J. Bact 1991, DOI: 10.1128/jb.173.21.6709-6713.1991; Smeal et al., Cell 1996, DOI: 10.1016/s0092-8674(00)81038-7).

2. While 2-3 replicates blots of most experiments are quantified, it appears that the claim that the degradation of FC20 and FC30 extends internally is just based on one blot (Supp. Fig. 1D). Replicates and quantifications should be added. It also appears that where replicate blots have been performed and quantified, only one blot is shown - ideally, all blots should be shown in the supplement.

For Fig. 1e-f and 2c, the original blots were added to the original Zenodo repository mentioned in the “resources availability” section. We have now included labeled versions of them as Supplementary Fig. 1d and Supplementary Fig. 2d, respectively. Supplementary Fig. 1d includes the three probes TRF of 6R, 23 kb internal probe of 6R and loading control. In addition, we have provided quantifications for the hybridization with the internal 6R region in Supplementary Fig. 1e :

3. The multiple references to oncogenesis in the abstract seem a stretch given the work's exclusive use of a single-celled microbe to study the effects of telomere shortening. I do not think that the use of the model systems in any way counts against the value of the work (in fact, its value lies in taking advantage of the systems' strengths). However, I feel the authors need to tone down their language to reflect the conclusions that their data directly support.

We regret any confusion that may have arisen. While we have retained essential references to cancer biology—particularly in discussing our novel hypothesis regarding the putative role of the shortest telomere—we have removed direct mentions of cancer in the following instances:

- **Abstract:**

The switch of the shortest telomere into dysfunction and subsequent processing in telomerase-negative cells thus serves as the mechanistic link between replicative senescence onset, genomic instability and the initiation of post-senescence survival, explaining the contradictory roles of replicative senescence in oncogenesis.

- **Discussion:**

The finding that the proliferative limit is set by the shortest telomere, and that genomic instability predominantly occurs at this telomere rather than impacting other chromosomes, could have significant implications for diagnosing predisposition to cancer and yet uncharacterized genetic disorders.

Reviewer #3 (Remarks to the Author)

Review Manuscript NCOMMS-25-35131-T

Summary:

In the manuscript entitled "Both Genome Instability and Replicative Senescence Stem from the Shortest Telomere in Telomerase-Negative Cells" Berardi P. et al combine yeast genetics with single-cell multifluiddic lineage tracing, modeling and whole-genome and telomere sequencing to address the consequences of one short telomere on genome stability and proliferation in *Saccharomyces cerevisiae*. The engineered FinalCut system allows the authors to investigate the cellular consequences of one short telomere of defined lengths at chromosome 6R in high resolution. This system is superior to previous telomere dysfunction models as it allows for precise temporal and telomere length control at a native subtelomeric context and thereby overcomes the intrinsic intracellular telomere heterogeneity. The analysis suggests that in budding yeast telomeres shorter than 30-40 bp become dysfunctional and cause replicative senescence and genome instability. Their work elegantly shows that one short dysfunctional telomere is sufficient to cause replicative senescence in budding yeast. Further, their data indicate that genome instability induced by one dysfunctional telomere acts predominantly in cis. To comprehensively address the impact of one critically short telomere on genome instability in trans further investigations will be needed.

Together, the work provides important mechanistic insights on how one single short telomere restricts eukaryotic proliferation capacity and causes genome instability. If the proposed model is conserved between other eukaryotes including human, the work has important clinical consequences for short telomere syndrome patients and Alternative Lengthening of Telomeres. In the future, it will be interesting to apply the developed methodology to investigate which other changes next to the telomere sequence loss occur at short telomeres and drive telomere dysfunction as well as the mechanisms that contribute to the observed genome instability.

We thank this reviewer for this positive comment.

Major comments:

1) The authors suggest that degradation of FC20 and FC30 may extend internally into the chromosome using a more internal TEL6R fragment. However, the representative blot shown in Supplementary Fig. 1d shows also reduced signal in the later time points of the FC70 sample. Further, what is the proposed mechanism for the loss of signal: nuclease-dependent degradation or terminal loss due to a GCR event? Can the authors please clarify this point?

We appreciate the reviewer's observation. Upon re-examination, we noted that the FC70 signal in the original Supplementary Fig. 1d was reduced at 9 hours, as was the loading control. To better evaluate degradation, we re-hybridized the blot—along with the other blots—using the internal probe for the 6R. The results of triplicate experiments are now presented in the revised Supplementary Fig. 1e (see above, reviewer #2 comment). Our findings confirm the disappearance of the signal for FC20 and FC30, with lesser losses observed above FC40. For FC50 and FC70, the losses are not significant.

2) In Fig. 2 b-c, the authors investigate the kinetics of telomeric signal loss at FinalCut Telomeres of varying length. Based on the representative blots the loss of signal seems to be dependent on the length of the FCx fragment with faster kinetics in FC40 relative to FC50. However, in the quantification, FC40 and FC50 have almost similar kinetics, especially in the first 2 h.

a. Can the authors please comment on this?

We thank this reviewer for spotting this. The graphs presented included technical replicates and the quantification method wasn't adapted to faint bands. We have now re-quantified one blot per biological independent experiment, in an equal manner for all of the blots, and in a manner adapted to a broader band intensities, that we present in a new Fig. 2c. The original blots are presented in Supplementary Fig. 2d. The conclusions in text remain valid, i.e., FC40 degrades faster than FC50.

New Fig2c:

b. Can the authors please specify why in some samples 4 or 5 individual data points are shown, even though the legend says that the average of three experiments is shown?

As explained above, the extra data points were incorrectly included as technical replicates. We have removed them to ensure only biological replicates are represented.

c. As telomere shortening due to the end replication problem is in large part replication-dependent, can the authors please provide the corresponding population doubling to the indicated time points?

We appreciate the reviewer's suggestion. The revised graph presented in the revised Fig2c now plots stability as a function of population doublings, which more accurately captures the biological phenomena discussed in this section.

3) The sequencing analysis of FC50 and FC70 revealed an average telomere shortening of 4.4 and 3.4 bp/PD, respectively (Fig. 3e). Can the authors please perform linear regression analysis and provide statistics using all three time points? Is there a biological reason why longer telomeres seem to shorten slower?

The numbers we have provided, 4.4 and 3.4 bp/PD of telomere shortening of FC50 and FC70, respectively, were obtained averaging the pool of the three experiments altogether for each time point. If we consider the lengths obtained in each of the three experiments, we obtain approximately the same results, 4.32 ± 1.06 and 3.46 ± 0.68 bp/PD. A student test gives a p-value of 0.301, showing that the shortening rates of FC50 and FC70 are not significantly different. We concluded that we have no reason, so far, to consider the hypothesis that FC50 and FC70 have a different shortening rates. We changed the section accordingly:

"We next examined telomeres in cells lacking telomerase. Sequencing of FC50 and FC70 over time shows gradual shortening of telomeres at an average (\pm SD) of 4.32 ± 1.06 and 3.46 ± 0.68 bp/PD, respectively. These rates are not statistically different (p-value=0.301), and are compatible with previous estimations"

4) Interestingly, the authors observed that 23.8% +/- 2.2 FinalCut telomeres shorten by 6 bp and end in position 51 (Fig. 3f), suggesting that those telomeres undergo telomere shortening prior to sampling. Is the shortening/observed telomere length at 0 hours dependent on when in the cell cycle the FinalCut was performed?

To clarify: at t=0h, cells have already been incubated in Galactose for 4 h (see Fig 3b). Under these conditions, telomeres have been undergoing cleavage for 4 h. As a result, at t=0 h, the FC populations already exhibits heterogeneity:

- some telomeres remain uncut (excluded in the telomere length measurements as they are easily identified by the presence of ttDNA in their sequence);
- most telomeres are cut,
- some telomeres are cut and have undergone one or two rounds of replication (the doubling time in Galactose being ~140 min, some telomeres could have completed up to two replication cycles).

The rationale of this experiment is that after Galactose replacement with glucose, and degradation of Cas9, the production of newly cut FC telomeres is stopped, so that we can trustily

monitor a population of telomeres after cut, and shortening rate can be measured as a variation in length over population doublings.

5) With increasing FC, the median PD before senescence onset increases (Fig. 4b and Supplementary Fig. 4). The author could calculate the telomere shortening rate per PD by plotting FC length against the median PD before senescence onset and performing linear regression analysis. Assuming linear shortening how long would the estimated shortest telomere in yeast with heterogeneous telomere length that enter senescence after median 30 PDs be? How does the length of the predicted shortest telomere relate to individual telomere length measurements in wild-type yeast?

As suggested by the reviewer, we plotted the FC length against the median PD before senescence onset and performed linear regression analysis as in the next figure. We find that the length of the shortest telomere in wild-type cells, compatible with a median lifespan of 30 generations is 164 bp. This is very similar to median length of the shortest telomere in isogenic strains without FC (we remind that these are *pTetO2-TLC1* strains, displaying shorter telomeres than *TLC1*), as found in Rat et al., Nat Comm 2025 – see Fig5a therein.

6) Can the authors please change their color code for some of their figures to allow better readability for people with color blindness (e.g. change red-green to red-blue in Fig. 4 and Supplementary Fig. 4)?

We thank the reviewer for this suggestion. We changed the colors to green-magenta as recommended by the American Society for Cell Biology.

7) To test whether short telomeres of 30-40 bp identified with FinalCut can also be detected at native telomeres the authors make use of a *PRNR3-TLC1* strain and sequence individual 6R telomeres. Chromosome arm-specific telomere length sequencing of Sholes SL et al (PMID: 34702734) suggests that the telomere length at chromosome 6R is quite average. This would predict that the shorter telomeres at other chromosome arms trigger the checkpoint response and consequently telomerase activity earlier. Why have the authors sequenced 6R telomeres and not for example the overall shorter 1L or 11L telomeres, so that the checkpoint activity can be more directly connected to the short telomere? Which outcome do the authors expect in case of sequencing the chromosome arms with the overall shorter telomeres?

One of the reasons we chose the 6R telomere is that it is indeed “quite average”. Importantly, the chromosome-end specific differences seen in a wild-type strain is unlikely to be the same in the *PRNR3-TLC1* strain, where telomerase is only activated when a critically short telomere activates the DNA damage checkpoint. The first critically short telomere might be, for example, the 1L telomere, but once extended, it will no longer be the shortest telomere. Since we are sequencing the telomeres from a

population of cells after many generations of growth, we do not expect the outcome to be any different had we instead sequenced the 1L or any other telomere.

8) The authors conclude based on their CAN1/GCR fluctuation assays that the genome instability during replicative senescence occurs primarily in cis (Fig. 7, Supplementary Fig 7 and Supplementary Table 9). This is in line with the presented data and it is intuitive that the arm with the shortest telomere is the one that is most genetically unstable. However, I have several comments and questions:

a. Why are the authors using sequential drug treatment and not simultaneous canavanine and 5-FOA selection to investigate GCR?

We realized that measuring the CAN1 mutation rate already provided us with a significant increase, and enabled the identification of possible BIR events by replica-plating on 5FOA, as suggested by Greider's previous work. That said, we recognize the value of this suggestion and have included new data using simultaneous selection (see below) to further address this point.

I have two remarks here: First, the additional growth time may result in more double-negative cells.

Indeed, we cannot at all exclude ongoing genomic instability, even though the canavanine-containing plates lack doxycycline, which allows telomerase reactivation and potential repair of broken chromosomes, thereby limiting further instability.

Second, CAN1 (and URA3) can be inactivated by several mechanisms (base substitutions, insertions/deletions, GCRs,...). In wild-type budding yeast CAN1 is mainly inactivated by base substitutions and smaller INDELS (PMID: 9702187, 12972632, 20961955), and infrequently by GCRs. So, if the CAN1 mutation rate (~10⁻⁷, mainly driven by base substitutions and INDELS) is orders of magnitude higher than the GCR rate (~10⁻¹⁰ for classical GCR assay, PMID: 11239397), it is not surprising that few GCR are detected in trans in the canavanine resistant colonies. The GCR assay in cis detects GCRs because the GCR rate is increased due to the adjacent critically short telomere. Consequently, the increased GCR rate in cis is comparable or higher than the CAN1 mutation rate and GCR are detected with sequential drug treatment. Importantly, it may be that the critically short telomere at 6R causes an increase in GCR rate in trans, even though at lower levels than in cis, but this increase is "masked" by the more frequent CAN1 inactivation events and the assay setup. Can the authors please measure GCR rates with simultaneous canavanine and 5-FOA selection?

We thank the reviewer for this insightful question. It is indeed plausible that additional chromosome ends are affected, particularly if the FC – potentially fusion-competent – telomere fuses with others. Telomere–telomere fusions and fusion–bridge–breakage (BFB) cycles have been documented in budding yeast telomerase-negative cells (Hackett et al. 2001, DOI: 10.1016/s0092-8674(01)00457-3; Chan & Blackburn 2003, DOI: 10.1016/s1097-2765(03)00174-6; Putnam & Kolodner 2017 DOI: 10.1534/genetics.112.145805). These observations have established that uncapped telomeres tend to undergo fusion events. Notably, such telomere fusions often lack telomeric repeats, or contain very few telomeric repeats, indicating that participating chromosome ends are mostly deprotected at the time of fusion. However, both the Hackett et al. 2001, DOI: 10.1016/s0092-8674(01)00457-3 and Hackett & Greider MCB 2003 DOI: 10.1128/MCB.23.23.8450-8461.2003, also provides evidence for end-resection being the major cause of genomic instability in the absence of telomerase, possibly overcoming the BFB cycles.

Building on these possibly contradictory observations, our study sought to advance understanding of this process by investigating the earliest stages, specifically by testing whether the shortest telomere initiates genomic instability. We focused on the point at which a single telomere approaches the critical length, prior to the widespread telomere uncapping or onset of major genomic instability. FC70 induces arrest after approximately 10 population doublings (~24 hours in galactose), which likely reflects a stage where only one telomere is critically short while the remaining telomeres are still capped. Our central question was whether this first critically short telomere already represents a threat to genome integrity and can initiate early cell cycle arrest or rearrangements, regardless of the specific mechanism involved, (e.g., NHEJ, end resection).

This said, this reviewer is right in saying that the double selection would indeed enable the direct scoring of GCRs, and, certainly, enable the evaluation of potential variations in the frequency of the rare events, especially in the trans conditions. We thus executed the experiments to find that in trans, there is no statistical difference in GCRs when a short telomere is generated in telomerase-negative conditions (~2-fold). This result, is now included as a new figures 6e, Supplementary Fig. 6g and discussed in the main text (see below).

In the maintext :

The basal GCR rate linked to simultaneous loss of CAN1-URA3 in classical GCR fluctuation assays is described orders of magnitude below the CAN1 mutation rate in telomerase negative mutants, prior to senescence onset⁶². Therefore, to better evaluate the increase in GCRs due to FC70 on GCR levels, we measured the GCR rate by simultaneous plating on canavanine and 5FOA (Supplementary Fig. 6g). Our results showed that the trans strain loses the 5L chromosome end at a basal rate of $\sim 1 \times 10^{-10}$, approximately 1,000-fold less frequently than the CAN1 mutation rate—consistent with published data in a similar conditions⁶². Notably, we detected no significant change upon FinalCut induction (Fig. 6e and Supplementary Tables 11-12). We concluded that a very short telomere in the 6R has no noticeable effect on the stability of another chromosome in trans. Yet, we cannot exclude the possibility of effects in trans, that would be more difficult to detect, such that telomere-to-telomere fusions and breakage-fusion-bridge cycles^{59,63}. In contrast—and consistent with the results from sequential plating on canavanine followed by 5-FOA—we observed an ~18-fold increase in GCRs in cis to FC70 (Fig. 6f, Supplementary Tables 11–12).

Taken together, these findings demonstrate that the shortest telomere in a telomerase-negative cell is the most prone to initiating genomic instability, which most frequently involves the adjacent chromosome arm.

b. Based on the presented sequencing of GCR+ colonies resulting in loss of the CAN1-URA3 cassette upon FinalCut in can1Δ cells, the authors conclude that non-reciprocal translocations are the signature of shortest telomeres. The authors could modify their GCR strain like Putnam et al. (PMID: 19641493) and integrate an antibiotic cassette telomeric to the introduced CAN1-URA3 cassette on 6R. This way they could simultaneously measure GCRs on canavanine-5-FOA containing media and quantify how many GCR events are indeed due to the chromosome arm loss.

In the study cited by this reviewer, Putnam et al. (Nature, 2009), the use of a GCR assay similar to our cis construct—where a homology region is present at the breakpoint—showed that 100% of Can^R-5FOA^R clones had lost the antibiotic resistance locus. This suggests that loss of the CAN1-URA cassette is systematically associated with loss of the entire chromosome arm in a context similar to ours. Also, a large majority of clones analyzed so far in the telomerase-negative context, do present a strong correlation between CAN1-URA3 status and chromosome arm presence or loss (this work, as well as Coutelier et al. 2018, Hackett 2001, 2003). Nevertheless, rare cases in which Can^R-5FOA^R would not correspond to the chromosome arm loss could be investigated. However, for the purposes of this study, we decided to focus on the predominant events—specifically, the non-reciprocal translocations in cis involving the shortest telomere.

c. As the authors observe stimulated by the homology of HXK1 and HXK2 non-reciprocal translocation as a prime GCR event. It will be interesting to delete either HXK1 or HXK2 and perform GCR assays (with simultaneous canavanine and 5-FOA treatment) in cis and trans. The absence of the homology site may result in more complex GCRs and increased in trans genome instability.

We appreciate the reviewer's suggestion and understand the rationale behind the proposed experiment. Deleting the native subtelomeric regions—naturally repetitive sequences—located between the CAN1-URA3 reporters and the last essential gene would likely increase cellular mortality and induce atypical forms of genomic instability that could indeed be of interest. However, we believe this would create an artificial context in which the selection for viable rearrangements becomes highly specific to the engineered configuration. Such a setup falls outside the more physiological framework that our study aims to model, and we have therefore not pursued this experiment in the present work.

d. The authors speculate that Exo1, Rad1 and BIR are involved in the formation of FinalCut-induced GCRs. To clarify the mechanism, can the authors please measure GCR rates in their FinalCut can1Δ GCR strain in the absence of Exo1 and Rad1, as well as in strains lacking Rad52, Rad51, Rad59 and Pol32 combination thereof?

As previously mentioned and suggested by this reviewer, we performed the GCR assay using direct double selection with canavanine and 5FOA. Both the trans and cis constructs were tested. For the cis construct, we specifically analyzed relevant mutants such as pol32, exo1, and rad1. However, rad51 and rad52 exhibit an excessively high baseline mortality rate, leading to rapid replicative senescence in the absence of telomerase. This phenotype effectively precludes meaningful assessment of the effects of telomerase inactivation, prior to cell death (discussed in Rat et al., 2025). A thorough investigation of these mutants and combinations of them would

necessitate the development of conditional alleles, which we consider beyond the scope of the current study. As for *rad59* we estimate it's beyond the scope of this work.

We added the new results in a new panel and added a new section in the maintext :

In the main text:

GCRs in cis of the shortest telomere in telomerase-negative cells depend on Pol32

*The non-reciprocal translocations detected in the cis GCR assay resemble BIR products. Notably, whole genome sequencing revealed translocation breakpoints at native subtelomeric homology regions. To directly test the BIR mechanism, we deleted POL32, which encodes a non-essential subunit of DNA polymerase δ required for BIR^{16,64,65}. In strains carrying the CAN1-URA3 marker in cis of FC70—positioned 5 kb from the last essential gene—and *can1-Δ*, POL32 deletion reduced GCR rates by 32-fold under induced FC70 conditions and 56-fold in its absence, compared to the POL32 wild-type context. This suppression of GCRs in *pol32-Δ* strongly supports BIR as the primary mechanism driving GCR formation in the absence of telomerase, when subtelomeric homology regions are present. Taken together our results support a model in which the absence of telomerase, combined with the presence of homology regions, drives genomic instability through BIR, prior to, or at least independently of, post-senescence survival, which also relies on BIR.*

*Given that dysfunctional FinalCut telomeres undergo substantial degradation (Fig. 1f, Supplementary Fig. 1d-e), we hypothesized that this end degradation could trigger BIR involving internal subtelomeric elements. Accordingly, both Exo1 exonuclease and Rad1 endonuclease have been implicated in increased genomic instability in telomerase-negative populations⁶⁰. However, neither Exo1 nor Rad1 is strictly required for BIR^{66,67}. While Exo1 mediates 5' to 3' resection of DNA double-strand breaks, generating single-stranded DNA for Rad51 filament formation and strand invasion, deletion of EXO1 paradoxically increases BIR frequency⁶⁸. Similarly, GCR rates in the classical GCR assay—which lacks homology between the last essential gene and the GCR reporters—are elevated in *exo1-Δ* strains, compared to EXO1⁶⁹. On the other hand, Rad1, though involved in processing non-homologous tails during homology-directed repair events, does not play a major role in the core BIR pathway itself⁶⁶. To solve this paradox, we tested the involvement of these nucleases in the GCRs in cis to FC70. We introduced *exo1-Δ* or *rad1-Δ* mutations into our cis GCR assay strain (Fig. 6b). We found that both mutations had only modest effects on GCR rates compared to wild type, with changes typically less than 3-fold, which is generally considered biologically insignificant in this context⁷⁰ (Fig. 6f). This suggests that Exo1 and Rad1 are not essential for BIR-mediated GCR formation in*

this context. Alternatively, EXO1 or RAD1 mutations do actually decrease BIR-mediated GCR rate in the absence of telomerase, but other repair mechanisms may be upregulated in the absence of appropriate end degradation, resulting in an apparent unchanged GCR rate.

e. The authors conclude that the shortest telomere is causing genome instability primarily in cis. This interpretation is based mostly on measured CAN1/GCR rates with the CAN1-URA3 cassette integrated either at chromosome arm 5L or 6R. While being a powerful assay the GCR assay only scores for quite specific events – loss of the CAN1-URA3 cassette and sufficiently protected chromosome ends and growth so that the event can be counted as a colony. For a critically short 6R telomere to cause an increased GCR rate in trans at 5L, the 6R arm would likely need to fuse with 5L, the fused chromosomes break resulting in loss of the CAN1-URA3 cassette and the broken ends stabilized by telomeres. This is a much more complex chain of events than a GCR in cis. Further, the critically short 6R telomere could in principle fuse with 5L and 30 other chromosome ends. This alone would suggest that the GCR rate at 5L is at least approximately 30x lower than the GCR rate than at 6R (assuming that all chromosome arms are equally fusion-prone). Thus, to investigate the possible genome instability impact of one critically short telomere at 6R in trans the authors could do two experiments:

i. Measure GCR rates (see above 8a-c).

Measurement of GCRs in trans showed no significant increase in Can^R 5FOA^R rates after the generation of a short telomere in telomerase-negative cells (New Fig. 6e). As the reviewer pointed out, this trans assay likely captures only a fraction of potential fusions and associated genomic instability, suggesting that the actual rate of such events may remain below our detection threshold.

ii. Measure 6R chromosome fusions with other chromosome arms in a fusion PCR assay like assays performed in mammalian cells by the Baird lab (PMID: 17908935).

We agree that this experiment would be worth pursuing, as we do expect to see some increase upon induction of FinalCut. However, since we did not observe a significant rise in GCR rates in trans using the fluctuation assay, which would be indicative of potential BFB cycles, we chose not to explore this direction further in the current study.

At the moment, we are developing new methods to analyse the whole-genome sequencing on populations of cells undergoing replicative senescence with FC70—without relying on the CAN1-URA3 selection or long-term viability. Our goal is to identify the spectrum of genome rearrangements that arise at the onset of senescence and to compare their relative frequencies.

9) The possibility that one single short telomere is necessary and sufficient also in other eukaryotes is an exciting hypothesis. One major difference between the used budding yeast strains and most other eukaryotic systems is, that the used yeast strains are haploid, whereas most eukaryotic systems are diploid. It will be interesting to investigate heterozygous or homozygous FinalCut diploid yeast and the impact of critical short telomere(s) in diploid yeast.

Diploids homozygous for mutations in telomerase components exhibit only a modest senescence phenotype, making them unsuitable for direct experimentation in this context (Meyer & Bailis 2008, DOI: 10.1111/j.1365-2958.2008.06353.x; Lowell et al., 2003, DOI: 10.1093/genetics/164.3.909). While we attempted to establish the assay in pseudodiploids, further optimization is required to fully characterize this system. We therefore consider this approach beyond the scope of the current

study.

Minor comments:

1) Page 8 line 220-221: please add *SacI* site and I am wondering whether the “a common stretch of degenerated TG1-3 repeats”, is redundant to “followed by TG1-3 of variable length and divergent sequence among the telomere molecules”?

We have now included a reference to the *Sac I* restriction site in the text.

Regarding the potential redundancy, there is indeed a distinction between the two types of telomeric repeats. As the reviewer can see in Supplementary Fig. 2b, immediately following the *ttDNA*, all telomeres share an identical stretch of TG repeats. Sequence divergence only becomes apparent approximately 80 bp downstream. This pattern reflects the action of telomerase in budding yeast, which adds degenerate repeats. Consequently, the terminal regions of the telomeres contain TG repeats that can no longer be aligned. There are therefore two distinct “types” of telomeric repeats.

2) Page 8 line 228-229: In Supplementary Fig. 3c can you please indicate which of the telomeres experienced *de novo* telomere re-elongation?

As indicated in the figure legend, our intention was to display only re-elongated sequences. However, two sequences that were clearly not elongated were incorrectly classified and included in the figure. We have now removed these sequences to ensure that only *de novo* re-elongated sequences are shown. The legend now reflects this change. Coincidentally, the two removed sequences each came from the two separate independent experiments, so their exclusion does not affect the representativity of either experiment. We appreciate the reviewer’s careful attention in identifying this oversight.

3) Page 12 line 331: How is the “long” cell cycle threshold defined? Dependent on the media the ratio is 2x and 1.6 fold? Please explain.

As stated in Methods section, “the threshold for prolonged cell cycles was set to [mean+ 2xSD] of the dataset, the mean and SD being computed from cell cycle durations in media containing raffinose only or raffinose and galactose.”

This threshold was adapted from our previous study (Martin et al. 2023), where we used mean + 3xSD for growth in glucose. For raffinose or galactose, we adjusted the threshold to mean + 2xSD due to the higher variability observed in these conditions (i.e., larger standard deviations). This approach thus represents an extrapolation of our glucose-based methodology.

4) In Supplementary Fig. 3 b-f the authors could mark/highlight the regions like FC50, and the *ttDNA*, similar to Fig. 1a.

We have changed the figures according to this suggestion.

5) Can you please add as comparison Supplementary Fig. 4d to Fig. 4b? Can the author please also indicate in Fig. 4b and the Supplementary Fig. 4 panels the median cell divisions before senescence onset?

We have changed the figures according to this suggestion and added the median cell divisions before senescence onset.

6) In some figures the light blue/turquoise text is very light and difficult to read if printed (e.g Fig. 1, Fig. 4b FC20; Supplementary Fig. 1 FC0 and FC20; 5b FC0). Can you please change it to a color that is more visible?

We have changed the figures according to this suggestion.

7) For the mutation rates please check number of cultures (Supplementary Table 9). Please add some additional cultures as described in methods for the ones that have very few cultures.

The experiment shown in Supplementary Fig. 5e indeed only involves 5 independent cultures of one transformant. It is a variant of Supplementary Fig 5f and Fig. 6d and can be removed without changing the message of the manuscript. We have instead chosen to add a note in Table 10 (former Table 9) to explain this. Methods were changed accordingly.

Five independent transformants with pT126 plasmid of relevant strains (Supplementary Table 1)

...

For each set of experiments, at least two independent original transformants for the FinalCut constructs were involved unless otherwise stated in triplicate experiments (each involving 5 transformants of pT126).

8) Supplementary Fig. 6 is missing. Please adjust the numbering of Supplementary Figures accordingly.

We have modified according to this suggestion

9) Please use either "." or ";" throughout the manuscript e.g. Supplementary Table 5 pH7.2 or pH7,4.

We sincerely apologize for these oversights and appreciate your attention to detail. All errors have now been corrected.

Reviewer #1 (Remarks to the Author):

I am satisfied that the revised manuscript addressed my comments (although I would like the authors to still clarify in the new figure 4b what the different shades of green exactly represent).

We are grateful for reviewer's remark and the time invested in reviewing our work.

As for Fig4b we changed the legend as follows (includes Fig4a):

a, Microfluidics experimental setting. FinalCut strains are introduced in a microfluidics circuit. Cells are allowed to invade microcavities through cell divisions. Carbon source is then changed to raffinose and doxycycline is added to inactivate telomerase up to the end the experiment. Galactose is next added for 6 hours to induce Cas9. Durations between two cell divisions are recorded, for consecutive cell divisions up to cell death. Variations in cell cycle durations are illustrated by the colour gradient, ranging from green (short) to magenta (long).

b, Description of the two routes to senescence found in microfluidics experiments of control telomerase-negative cells³⁴. Type A cells undergo senescence in a single abrupt transition from normal cell cycles to terminally prolonged cell cycles (colours as in a). Type B cells undergo series of prolonged cell cycles, followed by normal cell cycles, before ending their route by terminally abnormally long cell cycles.”

Reviewer #2 (Remarks to the Author):

The authors have improved the manuscript through the revision process, and I am happy they have adequately dealt with the questions/points that I raised.

We thank the reviewer for his/her positive feedback and the careful attention given to our manuscript.

Reviewer #3 (Remarks to the Author):

My main comments have been addressed in the revised manuscript of this interesting work. I especially appreciate the author's work on GCR fluctuations and pol32/exo1/rad1 knock-out strains.

We sincerely appreciate the reviewer's feedback and the time dedicated to evaluating our manuscript.